# TUI: A Conformal Uncertainty Indicator for Continual Test-Time Adaptation

## Abstract

Continual Test-Time Adaptation (CTTA) addresses the challenge of adapting models to sequentially changing domains during the testing phase. Since no ground truth labels are provided, existing CTTA methods rely on pseudo-labels for self-adaptation. However, CTTA is prone to error accumulation, where incorrect pseudo-labels can negatively impact subsequent model updates. Critically, during testing, a CTTA method cannot detect its mistakes, which may then propagate through further adaptations. In this paper, we propose a simple uncertainty indicator called TUI for the CTTA task based on Conformal Prediction (CP), which generates a set of possible labels for each test sample, ensuring that the true label is included within this set with a given coverage probability. Specifically, since domain shifts can undermine the coverage of predictions, making uncertainty estimation less dependable, we propose compensating for coverage by dynamically measuring the domain difference between the target and source domains in continuously changing environments. Moreover, after estimating uncertainty, we separate reliable test pseudo-labels and use them to discriminatively enhance the adaptation process. Empirical results demonstrate that our algorithm effectively estimates the uncertainty for CTTA under a specified coverage probability and improves adaptation performance across various existing CTTA methods.

## 1 Introduction

Recently, Continual Test-Time Adaptation (CTTA) (Wang et al., 2022) has garnered significant attention for its ability to enable trained models to handle various unknown test domain shifts through self-adaptation. This innovative approach aims to enhance model robustness and adaptability during the testing phase, addressing the dynamic nature of real-world data, such as autonomous driving (Sójka et al., 2023) and medical imagining (Chen et al., 2024). However, a critical challenge arises in many testing scenarios where the cost of incorrect predictions is prohibitively high. When self-adaptation is based on unreliable predictions, it may lead to severe error accumulation, compromising the model's performance. Therefore, effectively measuring the uncertainty of model outputs becomes crucial to mitigate losses and allow for human intervention or termination.

Some uncertainty estimation methods are based on Bayes rule, such as Bayes approximation (Maddox et al., 2019) and Monte Carlo dropout (Gal & Ghahramani, 2016), requiring high computational complexity or rely on model selection, thus difficult to be applied to testing time. Moreover, some methods directly use the output logits to form uncertainties such as entropy (Shi et al., 2024), which may suffer from confidence dilemma that unreliable logits give unreliable uncertainty estimations. In contrast, Conformal Prediction (CP) (Vovk et al., 2005) offers a promising solution for measuring uncertainty in predictions, which produces set-valued predictions that serves as a wrapper around existing models. CP is with the following compelling advantages. First, CP is model-agnostic, which means it does not require any assumptions about the model, making it applicable to any pretrained model without necessitating modifications. Second, CP yields controllable coverage, which means CP allows the true label coverage probability to be pre-specified and ensuring that this probability is met. These advantages meet the scenario of CTTA that continuously measuring the output uncertainties for a pre-trained model in testing time without confidence dilemma issue.

However, incorporating CP into unsupervised CTTA presents significant challenges. Traditional CP requires the assumption of data exchangeability, which refers to the assumption that the order in

Figure 1: In the task of CTTA, a test sample $x$ may be drawn from a different distribution in a long term testing phase. Traditional methods rely on the self-adaptation based on the prediction and ignore the uncertainty may cause error accumulation. TUI provides a technique of uncertainty measurement based on CP. For the test sample, if TUI outputs a prediction set with small size ($> 0$), it is regarded as reliable and yielding a large loss weight in adaptation. Large prediction set means unreliable prediction. The coverage means that the true label is included in The example image is sampled from ImageNet (Deng et al., 2009).

which the data points are observed does not matter. The assumption is violated under domain shift conditions, thus leading to the coverage gap issue (Barber et al., 2023). The coverage gap means that the uncertainty estimation is under the coverage much less than the given expectation. That is, the uncertainty estimation is not trustworthy in this situation.

In this paper, we explore the feasibility of using CP in testing scenarios by addressing the coverage gap challenge and propose a simple uncertainty measurement method named Test Uncertainty Indicator (TUI). *The goal of TUI is to output the uncertainty of testing for each test example with a given trained model. The key motivation for TUI is to compensate the coverage gap when domain shifts and output reliable uncertainty level.* Specifically, following CP, TUI maintains a static source calibration set with labels in the pre-training phase. To evaluate the uncertainty for an example during testing time, TUI measures the domain shifts by considering both model and data differences. Then, a quantile for the test sample is calculated based on the calibration set, and the domain shift level is used to compensate for the quantile to achieve better coverage. Last, a non-conformity threshold is obtained by the compensated quantile and outputs the corresponding prediction set, where its size is treated as the indication of uncertainty level. Moreover, based on the CP results, we design a simple enhanced adaptation method on confident test samples, which can also be applied to existing CTTA methods. We find applying more adaptations on samples with reliable predictions will get good testing performance. As shown in Fig. 1, a traditional CTTA block consists of a point prediction and an adaptation, the proposed TUI provides the testing uncertainty and helps the adaptation. We evaluate on three benchmark datasets and find that the proposed TUI can better evaluate the test uncertainty than other CP methods. By integrating the CP-based adaptation strategy, existing methods achieve better reliability and robustness of model predictions in dynamic and uncertain test environments.

Our contributions are three-fold:

(1) We propose a simple uncertainty estimation method TUI for CTTA to measure the test uncertainty for each test prediction. TUI is model-agnostic and relies only on a small size of calibration set.

(2) We propose an adaptation method based on the TUI estimation, which enhances the reliable test adaptation.

(3) We evaluate our method on benchmark datasets and help multiple existing CTTA methods measure their test uncertainty and achieve better performance via our adaptation strategy.

## 2 RELATED WORK

### 2.1 CONTINUAL TEST-TIME ADAPTATION

Test-Time Adaptation (TTA) enables the model to dynamically adjust to the characteristics of the test data, i.e. target domain, in a source-free and online manner (Jain & Learned-Miller, 2011; Sun et al.,

2020; Wang et al., 2020). Recently, CTTA (Wang et al., 2022) has been introduced to tackle TTA within a continuously changing target domain, involving long-term adaptation. This configuration often grapples with the challenge of error accumulation (Tarvainen & Valpola, 2017; Wang et al., 2022). Specifically, prolonged exposure to unsupervised loss from unlabeled test data during long-term adaptation may result in significant error accumulation. Additionally, as the model is intent on learning new knowledge, it is prone to forgetting source knowledge, which poses challenges when accurately classifying test samples similar to the source distribution. To solve the two challenges, the majority of the existing methods focus on improving the confidence of the source model during the testing phase. These methods employ the mean-teacher architecture (Tarvainen & Valpola, 2017) to mitigate error accumulation, where the student learns to align with the teacher and the teacher updates via moving average with the student. As to the challenge of forgetting source knowledge, some methods adopt augmentation-averaged predictions (Wang et al., 2022; Brahma & Rai, 2023; Döbler et al., 2023; Yang et al., 2023) for the teacher model, strengthening the teacher's confidence to reduce the influence from highly out-of-distribution samples. Some methods, such as Döbler et al. (2023) and Chakrabarty et al. (2023a), propose to adopt the contrastive loss to maintain the already learnt semantic information. Some methods believe that the source model is more reliable, thus they are designed to restore the source parameters (Wang et al., 2022; Brahma & Rai, 2023). Though the above methods keep the model from confusion of vague pseudo labels, they may suffer from overly confident predictions that are less calibrated. To mitigate this issue, it is helpful to estimate the uncertainty in the neural network.

## 2.2 Conformal Prediction

CP was first introduced in Gammerman et al. (1998) as a method for quantifying uncertainty in both classification and regression tasks. Vovk et al. (2005) provides a formalized introduction to conformal prediction as well as application (and associated theoretical results) in multiple data settings, e.g., online and batch procedures. Conformal prediction is a robust framework for quantifying uncertainty in machine learning models, especially in high-stakes applications where reliability is crucial. It provides a means to generate prediction sets that contain the true outcome with a specified probability, without relying on assumptions about the underlying data distribution. CP was pioneered by Vladimir Vovk and colleagues in the 1990s, focusing on the concept of exchangeability and the use of nonconformity scores (Vovk et al., 2005). The framework was further developed to include various modifications and extensions (Angelopoulos & Bates, 2021). The foundational book by Vovk et al. (2005) provides a comprehensive introduction to the theory and applications of conformal prediction, emphasizing its distribution-free nature. CP has been applied to a wide range of problems, including medical diagnostics (Caruana et al., 2015), autonomous vehicles (Lekeufack et al., 2023), and financial decision-making, where the quantification of uncertainty is critical for safety and trust. Researchers have extended conformal prediction to handle more complex scenarios, such as distribution shift (Tibshirani et al., 2019), distribution drift (Barber et al., 2023), and time-series data (Lei & Wasserman, 2014). CP has been coupled with risk control techniques to provide guarantees on various performance metrics, such as false discovery rate in multi-label classification (Farinhas et al., 2023). Recent work has explored the interplay between calibration techniques like temperature scaling and conformal prediction methods, revealing the impact of calibration on the performance of conformal predictors (Dabah & Tirer, 2024).

## 3 Preliminary: CTTA and Conformal Prediction

**Continual Test-Time Adaptation (CTTA).** Given a classification model pre-trained on a source domain, CTTA methods adapt the source model to the unlabeled target data, where the domain continuously changes. Because the adaptation is conducted during test time, which means the model needs to output the prediction immediately then update the model. The unsupervised dataset of target domains are denoted as $\mathcal{D}^k = \{x_m^k\}_{m=1}^{N^k}$, where $k$ is the target domain index. For each test sample, CTTA conducts two major operations including testing and adaptation. For testing, the model needs to output the prediction of the model. For adaptation, the model needs to adapt to the testing sample without ground-truth. Just because no label is given, a CTTA model is prone to error accumulation. To avoid this, a uncertainty estimation should be given for each testing samples. In this paper, we use conformal prediction to evaluate prediction uncertainties.

**Conformal Prediction (CP)** and **Coverage Gap Issue**. We first introduce CP under a multi-class classification task with total $K$ classes. Let $\mathcal{X}$ be the input space and $\mathcal{Y} := \{1, \cdots, K\}$ be the label space. We use $\pi : \mathcal{X} \to \mathbb{R}^K$ to denote the pre-trained neural network that is used to predict the label of a test sample. The model prediction in this classification tasks is generally made as

$$\hat{y} = \arg\max_{y \in \mathcal{Y}} \pi(y|x), \tag{1}$$

where $\pi(y|x)$ can be seen as the confidence of that $x$ being labeled to class $y$. To provide a uncertainty guarantee for the model performance, CP (Vovk et al., 2005) is designed to produce prediction sets containing true labels with a desired probability. Instead of only predicting point labels (only labels with max confidence will be selected) from the model outputs, standard conformal prediction takes a black-box prediction model, a calibration data set, and a new test example $x \in \mathcal{X}^{\text{test}}$ with unknown label $y \in \mathcal{Y}^{\text{test}}$, creating a prediction set $\mathcal{P}(x) \subseteq \mathcal{Y}^{\text{test}}$ and satisfies marginal coverage:

$$\mathbb{P}(y \in \mathcal{P}(x)) \geq 1 - \alpha, \tag{2}$$

for a coverage level $\alpha \in [0, 1]$ specified by the user. $\alpha$ is generally considered to represent a user pre-specified error rate. For instance, if $\alpha$ is set to $0.1$, the resulting prediction set is expected to achieve a $90\%$ coverage rate. In other words, there is a $90\%$ probability that the true label will be included within the prediction set.

However, the coverage in Eq. (2) is guaranteed only when the testing domains are with the same distribution with the training domain, say data exchangeability (Vovk et al., 2005; Barber et al., 2023; Zaffran et al., 2022; Bhatnagar et al., 2023; Gibbs & Candès, 2022; Farinhas et al., 2023; Zou & Liu, 2024). When domain shifts, the exchangeability is not satisfied, thus the coverage will significantly drop. As observed by Yilmaz & Heckel (2022) and Bhatnagar et al. (2023), even subtle shifts makes coverage drop from the desired $90\%$ to $60\%$ on Imagenet-Sketch dataset. This phenomenon is called Coverage Gap (Barber et al., 2023), which is defined as follows:

$$\kappa = (1 - \alpha) - \mathbb{P}\left\{y \in \mathcal{P}(x)\right\}, \tag{3}$$

where $1 - \alpha$ is the expected coverage and $\mathbb{P}\left\{y \in \mathcal{P}(x)\right\}$ is the obtained coverage. To fill in the coverage gap, NexCP (Barber et al., 2023) generalizes CP by employing weighted quantiles and a randomization technique, enabling robust predictive inference even when data exchangeability assumptions are violated. However, this method is designed for training phase and highly depends on a pre-defined domain shift value, which is not allowed in testing time. Moreover, Yilmaz & Heckel (2022) propose a QTC method to recalibrate the quantile for coverage compensation. However, QTC suffers from the unreliable domain gap measurement in continual domain shifts and ignore the model differences. More details about existing non-exchangeable CP can be found in Section 4.3.1.

Motivated by this, in this paper, we seek to design a CP method for CTTA to act as an uncertainty indicator during testing time, and solve the coverage gap issue. Moreover, we would present to improve the adaptation in CTTA via the uncertainty measurement.

## 4 TEST UNCERTAINTY INDICATOR FOR CTTA

### 4.1 CONFORMAL PREDICTION WITH QUANTILE COMPENSATION

In this section, we propose a simple uncertainty indicator based on CP for CTTA task named Test Uncertainty Indicator (TUI). TUI is based on CP, and the major challenge of TUI is the coverage gap when domain shifts as mentioned in the above section. In the following, we introduce how to build a simple uncertainty indicator for CTTA task step by step.

#### 4.1.1 STEP 1: PREPARING CALIBRATION SET

First, following Vovk et al. (2005), CP needs to build a calibration set to approximate the source distribution for efficient computation. We select a part of labeled source data as the calibration set in our implementation. In real-world applications, the calibration set is easy to obtain, such as split from the source training set or further collections, making sure the calibration set and the training data are drawn from a same distribution. We will discuss the storage of calibration set construction in the end of the section. Specifically, we denote the calibration set as $\mathcal{C} = \{(x_1, y_1) \cdots, (x_{|\mathcal{C}|}, y_{|\mathcal{C}|})\} \subset \mathcal{D}^{\text{val}}$. The calibration set should be built before test phase. Note that our method is only applied to CTTA tasks with this prepared calibration set, where the calibration data can be regarded to a fixed clue of training distribution.

### 4.1.2 STEP 2: COMPUTING JOINT DOMAIN SHIFTS

Existing non-exchangeable CP methods fail to estimate the continual domain shifts in CTTA, such as NexCP (Barber et al., 2023) and QTC (Yilmaz & Heckel, 2022). These methods either assume that the domain shift is known or ignore the issue of error accumulation in the model during CTTA. In many existing domain difference measure methods, they directly compute distribution distance based on the current model. For example, DSS (Chakrabarty et al., 2023b) uses the cosine distance between the prototypes of source domain and the current domain as the signal of domain shifts. However, because the error accumulation, the current model could be not convincing enough. That is, the prototypes may not represent the real data distributions. To this end, we propose to further consider the *model shift* when measure the domain shifts.

In our method, to estimate the domain shifts during continuous test time, we consider both model and data difference. For model difference, we use both the source model with parameter $\boldsymbol{\theta}^{\mathrm{src}}$ and the current model with parameter $\boldsymbol{\theta}^{\mathrm{crt}}$. For data difference, we use both the calibration set $\mathcal{C}$ and the current test data $\mathcal{B}$. Specifically, we construct a joint probability distribution of calibration data and test data from both source and current models. The joint probability distribution is computed by

$$p(x) = \mathrm{softmax}\left(\mathrm{concat}(\pi_{\boldsymbol{\theta}^{\mathrm{src}}}(x), \pi_{\boldsymbol{\theta}^{\mathrm{crt}}}(x))\right). \tag{4}$$

In this way, each sample can be represented by both the source and current models. Then, for the joint distribution difference measurement, we use

$$\rho = \frac{1}{|\mathcal{C}||\mathcal{B}|} \sum_{x^{\mathrm{calib}} \in \mathcal{C}} \sum_{x^{\mathrm{test}} \in \mathcal{B}} D_{\mathrm{JS}}(p(x^{\mathrm{test}})||p(x^{\mathrm{calib}})), \tag{5}$$

where $D_{\mathrm{JS}}$ is the Jensen-Shannon (JS) divergence, which is known as symmetric and stable. In the context of CTTA, comparing the distribution differences of joint feature representations from the source and current models, there are several advantages. First, joint feature representation captures correlations between different features, providing a more holistic view of the data distribution and how different models process it. Second, by combining multiple features, the joint distribution can better reflect subtle differences between domains, enhancing the precision of JS divergence measures. Last, comparing joint feature distributions allows for a more detailed assessment of how much the current model has gained compared to the source model.

### 4.1.3 STEP 3: COMPENSATING QUANTILE THRESHOLD

When obtaining the domain shift score $\rho$, we can compensate the coverage of CP in CTTA. Specifically, we use the threshold conformal predictor (THR, Sadinle et al. (2019)) to construct the prediction sets by thresholding output. In general, the prediction set for the test sample $x$, denoted as $\mathcal{P}(x; \tau)$, are defined as the set of indices where the non-conformity score are greater than or equal to a threshold value $\tau$. In traditional CP, the threshold value $\tau$ is determined as the $(1 - \alpha)(\frac{|\mathcal{C}|+1}{|\mathcal{C}|})$-quantile of the calibrated non-conformity scores, as computed as follows:

$$\tau^* = \mathrm{Quantile}(\mathcal{C}, (1 - \alpha)) = \inf\left\{\tau : \frac{1}{|\mathcal{C}|}\sum_{x \in \mathcal{C}} \mathbb{I}_{\{s(\pi(x)) < \tau\}} \geq \frac{|\mathcal{C}| + 1}{|\mathcal{C}|}(1 - \alpha)\right\}, \tag{6}$$

where the non-conformity scores $s(\cdot)$ represent the threshold required for each calibration example to achieve coverage, and can be easily computed by one minus the softmax output of the true class:

$$s(\pi(x)) = 1 - \hat{y}. \tag{7}$$

Finally, we compensate the threshold based on the computed domain shift estimation $\rho$ in Eq. (5):

$$\hat{\tau} = \tau^* - \beta \cdot \rho, \tag{8}$$

where $\beta$ is a predefined factor. The compensation can be seen to include some more uncertain classes to the prediction set to meet the coverage requirement.

### 4.1.4 STEP 4: COMPUTING THE PREDICTION SET.

For the test sample $x$, we can compute the corresponding prediction set by thresholding

$$\mathcal{P}(x; \hat{\tau}) = \{y | s(y|\pi(x)) < \hat{\tau}, \forall y \in \mathcal{Y}\}, \tag{9}$$

where $\mathcal{Y}$ is the label space. The size of the prediction set can be seen as the measurement of uncertainty. Generally, a prediction set with large size is regarded as uncertainty. The TUI algorithm can be seen in Algorithm 1.

---

**Algorithm 1** Test Uncertainty Indicator in CTTA

---

**Input:** Test data point $x$, Pre-trained model $\pi$, calibration set $\mathcal{C}$, test data stream $\mathcal{X}^{\text{test}}$
  1: Point prediction via the pre-tained model: $\hat{y} = \arg\max_{y \in \mathcal{Y}} \pi(y|x)$
  2: Measure domain difference $\rho$ using Eq. (5)
  3: Compute non-conformity scores for calibration set using Eq. (7)
  4: Obtain the threshold $\tau^* = \text{Quantile}(\mathcal{C}, 1 - \alpha)$
  5: Compensate threshold via $\hat{\tau} = \tau^* - \beta \cdot \rho$
  6: Set prediction via threshold: $\mathcal{P}(x; \hat{\tau}) = \{y | s(y|\pi(x)) < \hat{\tau}, \forall y \in \mathcal{Y}\}$
**Output:** Point prediction $\hat{y}$, Set prediction $\mathcal{P}$

---

## 4.2 TUI-GUIDED ADAPTATION

Now we have the prediction set for each test sample, and the size of the prediction set represents the uncertainty level of the prediction. In general, the set size is close to 1 but larger than 0 can be regarded to reliable. However, traditional CP methods focus on detecting violations of the exchangeability assumption rather than adapting to such changes (Fedorova et al., 2012; Volkhonskiy et al., 2017; Vovk et al., 2020). In the context of CTTA, we prefer to further improve the adaptations via the guidance from CP.

Motivated by this, we design a simple adaptation strategy for CTTA based on TUI, weighting the adaptation of each test sample according to its uncertainty. A test sample with more reliable prediction will be set to larger weight for adaptation. Taking the adaptation in Mean-Teacher-based methods (Wang et al., 2011; Brahma & Rai, 2023; Döbler et al., 2023) as an example, these methods construct a mean-teacher structure based on the source pretrained model. The mean-teacher structure contains a student model and a teacher model, where the student updates via learning logits from the teacher, and the teacher then updates via exponential moving averaging from the updated student. In this case, the TUI-guided adaptation on the student model can be represented by:

$$L = -\frac{1}{|\mathcal{B}|} \sum_{x \in \mathcal{B}} \gamma\left[x, \mathcal{P}(\mathcal{B}; \tau)\right] \cdot \hat{\pi}(x) \log \pi(x), \tag{10}$$

where $\hat{\pi}$ and $\pi$ are the teacher and student models, respectively. $\gamma\left[x, \mathcal{P}(\mathcal{B}; \tau)\right]$ is a function to assign weight to each adaptation and is highly related to the prediction set size:

$$\gamma\left[x, \mathcal{P}(\mathcal{B}; \tau)\right] = \begin{cases} \dfrac{\max_{x' \in \mathcal{B}}(|\mathcal{P}(x')|) - |\mathcal{P}(x)|}{\max_{x' \in \mathcal{B}}(|\mathcal{P}(x')|) - 1 + \delta}, & \text{if } |\mathcal{P}(x)| > 0, \\ 0, & \text{if } |\mathcal{P}(x)| = 0, \end{cases} \tag{11}$$

where $\mathcal{P}(x) = \mathcal{P}(x; \tau)$ for simplicity and $\delta$ is a minimum value (like $1e - 9$) to avoid zero denominator. Eq. (11) gives a simple relative weight for a mini-batch adaptation. Note that if the prediction set size is 1, *i.e.*, $|\mathcal{P}(x)| = 1$, we have $\gamma \approx 1$ (if the max prediction set size is larger than 1, $\gamma = 1$), which is considered as the most reliable. Moreover, if $|\mathcal{P}(x)| = 0$, that means the an empty prediction set, we set the most unreliable prediction across the mini-batch. If the batch size is 1 for strict online setting, one will obtained binary weights.

## 4.3 DISCUSSION

### 4.3.1 COMPARISON WITH EXISTING NON-EXCHANGEABLE CP METHODS

We compare our TUI with two recent non-exchangeable CP methods, including NexCP (Farinhas et al., 2023) and QTC (Yilmaz & Heckel, 2022). First, both NexCP and QTC are designed only for uncertainty indication instead of adaptation improvement. NexCP is designed for training time, where it specifies a constant to represent the domain difference from the source domain to the target domain. Specifically, NexCP directly compensates the coverage by

$$\mathbb{P}(y \in \mathcal{P}(x)) \geq 1 - \alpha - 2\sum_{i=1}^{n} \tilde{w}_i \epsilon_i, \tag{12}$$

where $\epsilon_i$ is a predefined constant measure of how much the distribution has shifted from the test sample to the $i$-th calibrated sample and $w_i$ is a corresponding weight. NexCP will satisfy marginal

Table 1: Results of combining TUI with exiting CTTA methods on CIFAR10-to-CIFAR10C. All results are evaluated with the largest corruption severity level 5 in an online fashion. For each SOTA method, the first line means the vanilla implementation only with TUI for uncertainty estimation, and the second line means the method uses uncertainty to guide the adaptation. The table details are the same in Tables 2 and 3.

| Method(TUI) + CPAda | $\alpha = 0.3$ | | | | | | $\alpha = 0.2$ | | | | | | $\alpha = 0.1$ | | | | | |
|---|---|---|---|---|---|---|---|---|---|---|---|---|---|---|---|---|---|---|
| | ERR | COV | INE | NLL | BS | ECE | ERR | COV | INE | NLL | BS | ECE | ERR | COV | INE | NLL | BS | ECE |
| Tent | 19.66 | 28.99 | 0.32 | 1.78 | 0.17 | 0.17 | 19.66 | 67.71 | 0.76 | 1.78 | 0.17 | 0.17 | 19.66 | 39.29 | 0.43 | 1.78 | 0.17 | 0.17 |
| + CPAda | 17.86 | 66.61 | 0.74 | 1.45 | 0.15 | 0.16 | 18.26 | 76.67 | 0.93 | 1.49 | 0.16 | 0.19 | 17.60 | 84.44 | 1.21 | 1.41 | 0.15 | 0.17 |
| CoTTA | 17.17 | 58.69 | 0.64 | 0.64 | 0.12 | 0.09 | 17.17 | 66.88 | 0.83 | 0.64 | 0.12 | 0.09 | 17.17 | 80.66 | 1.23 | 0.64 | 0.12 | 0.09 |
| + CPAda | 16.95 | 64.62 | 0.73 | 0.64 | 0.12 | 0.10 | 17.01 | 72.83 | 0.90 | 0.66 | 0.12 | 0.12 | 16.5 | 84.91 | 1.26 | 0.64 | 0.12 | 0.14 |
| SATA | 16.84 | 36.23 | 0.37 | 0.60 | 0.11 | 0.07 | 16.84 | 46.92 | 0.48 | 0.60 | 0.11 | 0.07 | 16.84 | 54.96 | 0.57 | 0.60 | 0.11 | 0.07 |
| + CPAda | 16.61 | 67.08 | 0.76 | 0.64 | 0.11 | 0.11 | 16.55 | 75.87 | 0.92 | 0.65 | 0.11 | 0.10 | 16.53 | 84.78 | 1.28 | 0.65 | 0.11 | 0.12 |
| RMT | 17.79 | 69.73 | 0.84 | 0.78 | 0.13 | 0.16 | 17.79 | 75.95 | 0.97 | 0.78 | 0.13 | 0.16 | 17.79 | 82.88 | 1.19 | 0.78 | 0.13 | 0.16 |
| + CPAda | 17.46 | 70.87 | 0.85 | 0.77 | 0.13 | 0.12 | 17.53 | 76.59 | 0.98 | 0.78 | 0.13 | 0.14 | 17.76 | 82.81 | 1.18 | 0.8 | 0.13 | 0.13 |
| C-CoTTA | 15.09 | 51.69 | 0.53 | 0.86 | 0.12 | 0.15 | 15.09 | 59.12 | 0.61 | 0.86 | 0.12 | 0.15 | 15.09 | 68.48 | 0.73 | 0.86 | 0.12 | 0.15 |
| + CPAda | 14.98 | 69.06 | 0.75 | 0.90 | 0.12 | 0.16 | 14.88 | 73.76 | 0.81 | 0.90 | 0.12 | 0.16 | 14.89 | 83.99 | 1.23 | 0.92 | 0.12 | 0.17 |

coverage, and are exact when the magnitude of the distribution shift is known, which is infeasible in test time. In contrast, TUI is designed for testing, and measure the distribution shifts adaptatively.

QTC proposes to replace the user-specified $\alpha$ to a new coverage level $\beta_{\mathrm{QTC}}$ calculated as

$$\beta_{\mathrm{QTC}} = \min\left(\frac{1}{|\mathcal{C}|}\sum_{x\in\mathcal{C}}\mathbb{I}_{\{s(\pi(x))<\mathrm{Quantile}(\mathcal{B},\alpha)\}}, 1 - \frac{1}{|\mathcal{B}|}\sum_{x\in\mathcal{B}}\mathbb{I}_{\{s(\pi(x))<\mathrm{Quantile}(\mathcal{C},1-\alpha)\}}\right). \quad (13)$$

Based on the current model $\pi$, QTC finds a threshold on the scores of the model on the unlabeled samples and predicts the coverage level by utilizing how the distribution of the scores changes across test distribution with respect to this threshold. However, QTC ignore the adaptation on continual domain shifts may suffer serious error accumulation, making the current model unreliable. This leads to the CP results being unreliable too. Instead, our TUI considers the error accumulation and evaluates domain shifts based on a joint distribution difference. More details are shown in Appendix A.

### 4.3.2 DATA STORAGE FOR CALIBRATION IN TESTING TIME

In our method, we explore the feasibility of using CP in testing scenarios with the aid of additional samples for calibration. That means the testing system needs to store extra data, yielding more storage requirements. In fact, this is a common practice in continual learning. Many continual learning (Rolnick et al., 2019; Van de Ven et al., 2020) methods store and retrain previous training examples to avoid catastrophic forgetting of past tasks, named replay strategy. In comparison with replay, the calibration set in TUI is not used for adaptation but calibration in testing time, and the calibration set will not be updated in our method.

Moreover, in many real-world applications, it is feasible to pre-store data in training or utilize data in adaptation. Practical approaches in real-world settings involve storing samples to improve testing outcomes. Some methods even utilize additional unlabeled samples to enhance training (Goldman & Zhou, 2000; Zhu & Goldberg, 2022). Some methods such as Tomani et al. (2021) and Rahimi et al. (2020) leverage post-hoc calibration to achieve better performance under domain drift scenarios by using validation or calibration sets. Thus, it is reasonable to leverage the calibration set in test time.

## 5 EXPERIMENT

### 5.1 EXPERIMENTAL SETTING

**Dataset**. In our experiments, we employ the CIFAR10C, CIFAR100C, and ImageNetC datasets as benchmarks to assess the robustness of classification models. Each dataset comprises 15 distinct types of corruption, each applied at five different levels of severity (from 1 to 5). These corruptions are systematically applied to test images from the original CIFAR10 and CIFAR100 datasets, as well as validation images from the original ImageNet dataset.

**Pretrained Model**. Following previous studies (Wang et al., 2020; 2022), we adopt pretrained WideResNet-28 (Zagoruyko & Komodakis, 2016) model for CIFAR10to-CIFAR10C, pretrained

Table 2: Results of combining TUI with exiting CTTA methods on CIFAR100-to-CIFAR100C.

| Method(TUI) + CPAda | $\alpha = 0.3$ | | | | | | $\alpha = 0.2$ | | | | | | $\alpha = 0.1$ | | | | | |
|---|---|---|---|---|---|---|---|---|---|---|---|---|---|---|---|---|---|---|
| | ERR | COV | INE | NLL | BS | ECE | ERR | COV | INE | NLL | BS | ECE | ERR | COV | INE | NLL | BS | ECE |
| Tent | 62.13 | 42.00 | 4.48 | 7.72 | 0.51 | 0.24 | 62.13 | 79.22 | 31.81 | 7.72 | 0.51 | 0.24 | 62.13 | 50.66 | 5.30 | 7.72 | 0.51 | 0.24 |
| + CPAda | 49.58 | 69.02 | 17.44 | 3.58 | 0.36 | 0.15 | 52.66 | 73.81 | 19.01 | 4.10 | 0.39 | 0.17 | 56.21 | 89.40 | 37.31 | 4.59 | 0.43 | 0.19 |
| CoTTA | 32.23 | 58.02 | 1.32 | 1.27 | 0.15 | 0.05 | 32.23 | 70.62 | 2.60 | 1.27 | 0.15 | 0.05 | 32.23 | 79.83 | 4.81 | 1.27 | 0.15 | 0.05 |
| + CPAda | 31.91 | 68.94 | 1.94 | 1.32 | 0.16 | 0.08 | 31.87 | 77.20 | 3.08 | 1.32 | 0.16 | 0.08 | 32.05 | 85.39 | 5.97 | 1.33 | 0.16 | 0.09 |
| SATA | 31.73 | 28.47 | 0.33 | 1.24 | 0.15 | 0.07 | 31.73 | 64.89 | 1.27 | 1.24 | 0.15 | 0.07 | 31.73 | 37.64 | 0.47 | 1.24 | 0.15 | 0.07 |
| + CPAda | 30.70 | 65.11 | 1.28 | 1.20 | 0.15 | 0.07 | 30.48 | 72.51 | 1.80 | 1.19 | 0.15 | 0.08 | 30.26 | 82.20 | 3.17 | 1.18 | 0.15 | 0.08 |
| RMT | 31.34 | 69.01 | 1.75 | 1.48 | 0.18 | 0.14 | 31.34 | 76.95 | 2.80 | 1.48 | 0.18 | 0.14 | 31.34 | 84.00 | 5.20 | 1.48 | 0.18 | 0.14 |
| + CPAda | 31.20 | 69.45 | 1.68 | 1.45 | 0.18 | 0.13 | 31.34 | 76.48 | 2.68 | 1.47 | 0.18 | 0.14 | 31.14 | 84.22 | 5.17 | 1.47 | 0.18 | 0.13 |
| C-CoTTA | 30.23 | 49.91 | 0.75 | 1.41 | 0.17 | 0.14 | 30.23 | 58.06 | 1.00 | 1.41 | 0.17 | 0.14 | 30.23 | 57.78 | 0.99 | 1.41 | 0.17 | 0.14 |
| + CPAda | 29.88 | 67.32 | 1.32 | 1.57 | 0.18 | 0.15 | 29.52 | 75.63 | 2.02 | 1.63 | 0.19 | 0.16 | 29.25 | 84.38 | 3.76 | 1.67 | 0.20 | 0.17 |

Table 3: Results of combining TUI with exiting CTTA methods on ImageNet-to-ImageNetC.

| Method(TUI) + CPAda | $\alpha = 0.3$ | | | | | | $\alpha = 0.2$ | | | | | | $\alpha = 0.1$ | | | | | |
|---|---|---|---|---|---|---|---|---|---|---|---|---|---|---|---|---|---|---|
| | ERR | COV | INE | NLL | BS | ECE | ERR | COV | INE | NLL | BS | ECE | ERR | COV | INE | NLL | BS | ECE |
| Tent | 62.69 | 17.32 | 0.32 | 3.26 | 0.17 | 0.13 | 62.69 | 27.62 | 0.78 | 3.26 | 0.17 | 0.13 | 62.69 | 42.39 | 2.42 | 3.26 | 0.17 | 0.13 |
| + CPAda | 62.50 | 69.26 | 47.89 | 3.24 | 0.17 | 0.13 | 62.53 | 74.19 | 43.25 | 3.24 | 0.17 | 0.13 | 62.60 | 88.71 | 164.5 | 3.25 | 0.17 | 0.13 |
| CoTTA | 65.88 | 24.62 | 0.72 | 3.44 | 0.16 | 0.09 | 65.88 | 45.35 | 4.54 | 3.44 | 0.16 | 0.09 | 65.88 | 33.97 | 1.79 | 3.44 | 0.16 | 0.09 |
| + CPAda | 65.35 | 62.99 | 26.53 | 3.41 | 0.15 | 0.09 | 65.11 | 75.75 | 68.48 | 3.39 | 0.15 | 0.08 | 65.37 | 83.89 | 119.22 | 3.41 | 0.15 | 0.08 |
| SATA | 62.95 | 9.89 | 0.14 | 3.24 | 0.16 | 0.08 | 62.95 | 43.70 | 19.04 | 3.24 | 0.16 | 0.08 | 62.95 | 15.91 | 0.28 | 3.24 | 0.16 | 0.08 |
| + CPAda | 62.45 | 64.82 | 95.32 | 3.20 | 0.16 | 0.07 | 62.03 | 77.89 | 151.14 | 3.23 | 0.16 | 0.07 | 62.26 | 84.43 | 194.65 | 3.24 | 0.16 | 0.07 |
| RMT | 60.21 | 49.78 | 4.77 | 3.12 | 0.18 | 0.13 | 60.21 | 58.47 | 9.38 | 3.12 | 0.18 | 0.13 | 60.21 | 66.76 | 18.46 | 3.12 | 0.18 | 0.13 |
| + CPAda | 59.81 | 65.60 | 17.28 | 4.58 | 0.30 | 0.13 | 59.90 | 78.62 | 54.64 | 3.08 | 0.17 | 0.13 | 59.92 | 89.27 | 131.17 | 3.09 | 0.17 | 0.12 |
| C-CoTTA | 60.24 | 42.49 | 2.77 | 3.29 | 0.22 | 0.21 | 60.24 | 65.97 | 12.43 | 3.29 | 0.22 | 0.21 | 60.24 | 52.54 | 5.82 | 3.29 | 0.22 | 0.21 |
| + CPAda | 59.75 | 66.31 | 13.07 | 3.31 | 0.22 | 0.21 | 59.87 | 77.01 | 30.66 | 3.30 | 0.22 | 0.21 | 59.69 | 88.64 | 93.08 | 3.25 | 0.21 | 0.20 |

ResNeXt-29 (Xie et al., 2017) for CIFAR100-to-CIFAR100C, and standard pretrained ResNet-50 (He et al., 2016) for ImageNet-to-ImagenetC. Similar to CoTTA, we update all the trainable parameters in all experiments. The augmentation number is set to 32 for all methods that use the augmentation strategy. For fair comparison, we conduct all experiments in a same environment.

**Evaluation Metric**: We use three kinds of metrics including testing performance, CP performance and uncertainty measure. We use $\hat{\mathcal{D}}$ to represent the testing data with labels. (1) For testing performance, we use the error rate following existing CTTA methods (Wang et al., 2022) and the small, the better. (2) For CP performance, we leverage coverage and inefficiency for joint evaluation:

$$\text{COV} := \mathbb{E}_{(x,y)\in\hat{\mathcal{D}}}\mathbb{I}\left(y \in \mathcal{P}(x)\right), \quad \text{INE} := \mathbb{E}_{x\in\hat{\mathcal{D}}}\left|C(x)\right|. \tag{14}$$

The coverage should near to the user expectation and the inefficiency should be small but larger than 0. (3) For uncertainty measure, we use Negative Log Likelihood (NLL), Brier Score (BS, Brier (1950)) and Expected Calibration Error (ECE, Naeini et al. (2015)):

$$\text{NLL} = -\mathbb{E}_{(x,y)\in\hat{\mathcal{D}}}\log(p(y|x)), \text{BS} = \mathbb{E}_{(x,y)\in\hat{\mathcal{D}}}\left(p(x) - 1(y)\right)^2, \text{ECE} = \sum_{i=1}^{10}\frac{|\mathcal{B}_i|}{|\hat{\mathcal{D}}|}\left|\text{acc}(\mathcal{B}_i) - \text{conf}(\mathcal{B}_i)\right|, \tag{15}$$

where $1(\cdot)$ means onehot. In ECE, we split samples to 10 bins by probability, and $\text{acc}(\mathcal{B}_i)$ means the bin accuracy and $\text{conf}(\mathcal{B}_i)$ is the mean confidence of the bin.

## 5.2 MAJOR RESULTS

TUI is a play-and-plug uncertainty indicator. To evaluate the effect of TUI, we select several well-known and state-of-the-art methods for comparison. TENT (Wang et al., 2020) updates via Shannon entropy for unlabeled test data. CoTTA (Wang et al., 2022) builds the MT structure and uses randomly restoring parameters to the source model. SATA (Chakrabarty et al., 2023a) modifies the batch-norm affine parameters using source anchoring-based self-distillation to ensure the model incorporates knowledge of newly encountered domains while avoiding catastrophic forgetting. RMT (Döbler et al., 2023) combines symmetric cross-entropy with contrastive learning in CTTA. C-CoTTA (Shi et al., 2024) proposes to adjust the directions of domain shift therefore to keep the discriminative ability. All compared methods adopt the same pre-trained model. For each selected method, we use the proposed TUI for uncertainty measurement, and based on this, we compare two results: one without adaptation and one using TUI guidance for domain adaptation. These two results are represented as adjacent rows in the table, such as "CoTTA" and "CoTTA + CPAda".

The results are shown in Tables 1, 2 and 3 for CIFAR10-to-CIFAR10C, CIFAR100-to-CIFAR100C and ImageNet-to-ImageNet10C, respectively. We set the total calibration set sizes to 50, 100 and

Table 4: Comparisons with other non-exchangeable CP methods on CIFAR100C.

| $\alpha$ | CP Method | w/o Adaptation | | | | | | w/ Adaptation | | | | | |
|---|---|---|---|---|---|---|---|---|---|---|---|---|---|
| | | ERR | COV | INE | NLL | BS | ECE | ERR | COV | INE | NLL | BS | ECE |
| | Baseline | 32.82 | 34.34 | 0.44 | 1.28 | 0.15 | 0.04 | - | - | - | - | - | - |
| 0.3 | THR (Sadinle et al., 2019) | 32.82 | 34.31 | 0.44 | 1.28 | 0.15 | 0.04 | 31.65 | 41.06 | 0.52 | 1.31 | 0.15 | 0.06 |
| | NexCP (Barber et al., 2023) | 32.82 | 36.43 | 0.48 | 1.28 | 0.15 | 0.04 | 31.90 | 40.67 | 0.52 | 1.32 | 0.15 | 0.06 |
| | QTC (Yilmaz & Heckel, 2022) | 32.82 | 58.09 | 1.11 | 1.28 | 0.15 | 0.04 | 30.42 | 58.90 | 0.91 | 1.30 | 0.15 | 0.08 |
| | TUI (Ours) | 32.82 | 69.41 | 2.01 | 1.28 | 0.15 | 0.04 | 29.26 | 80.70 | 2.57 | 1.34 | 0.17 | 0.11 |
| 0.2 | THR (Sadinle et al., 2019) | 32.82 | 42.32 | 0.60 | 1.28 | 0.15 | 0.04 | 31.46 | 49.46 | 0.69 | 1.33 | 0.15 | 0.07 |
| | NexCP (Barber et al., 2023) | 32.82 | 44.31 | 0.65 | 1.28 | 0.15 | 0.04 | 31.18 | 48.90 | 0.68 | 1.31 | 0.15 | 0.06 |
| | QTC (Yilmaz & Heckel, 2022) | 32.82 | 65.28 | 1.59 | 1.28 | 0.15 | 0.04 | 29.79 | 68.05 | 1.25 | 1.3 | 0.16 | 0.09 |
| | TUI (Ours) | 32.82 | 77.11 | 3.26 | 1.28 | 0.15 | 0.04 | 29.19 | 85.57 | 3.95 | 1.34 | 0.18 | 0.12 |
| 0.1 | THR (Sadinle et al., 2019) | 32.82 | 54.72 | 0.98 | 1.28 | 0.15 | 0.04 | 30.35 | 60.52 | 1.00 | 1.32 | 0.15 | 0.08 |
| | NexCP (Barber et al., 2023) | 32.82 | 54.92 | 1.00 | 1.28 | 0.15 | 0.04 | 30.49 | 59.96 | 0.98 | 1.32 | 0.16 | 0.08 |
| | QTC (Yilmaz & Heckel, 2022) | 32.82 | 75.35 | 3.06 | 1.28 | 0.15 | 0.04 | 29.29 | 74.42 | 1.64 | 1.32 | 0.17 | 0.10 |
| | TUI (Ours) | 32.82 | 87.41 | 7.80 | 1.28 | 0.15 | 0.04 | 29.15 | 89.37 | 6.11 | 1.33 | 0.17 | 0.12 |

Table 5: Data storage analysis ($\alpha = 0.2$) and comparison with replay strategy.

| Method | Total Storage | ERR | COV | INE | NLL | BS | ECE |
|---|---|---|---|---|---|---|---|
| Baseline | | 32.85 | 34.34 | 0.44 | 1.28 | 0.15 | 0.04 |
| Soure Replay | 100 | 32.76 | 7.13 | 0.07 | 1.27 | 0.15 | 0.05 |
| TUI+CPAda | | 29.77 | 70.24 | 1.46 | 1.32 | 0.16 | 0.09 |
| Soure Replay | 200 | 32.67 | 7.84 | 0.08 | 1.26 | 0.14 | 0.05 |
| TUI+CPAda | | 29.65 | 74.10 | 1.80 | 1.33 | 0.16 | 0.10 |
| Soure Replay | 300 | 32.21 | 24.40 | 0.27 | 1.24 | 0.14 | 0.04 |
| TUI+CPAda | | 29.65 | 74.45 | 1.82 | 1.34 | 0.16 | 0.10 |

500 for CIFAR10C, CIFAR100C and ImageNetC, respectively. We use three expected coverage factors $\alpha = 0.1, 0.2, 0.3$, which represents that the user would like $90\%, 80\%, 70\%$ coverage for the prediction. As shown in the tables' results among these methods, using TUI for uncertainty estimation reveals two notable issues with the original methods. First, the results demonstrate good coverage, but the inefficiency is relatively high, indicating that TUI estimates a high level of uncertainty for these results. Second, while the results show excellent inefficiency, the coverage is low, suggesting that the model is overly confident in its predictions, which significantly deviate from the correct outcomes.

After employing the TUI-guided domain adaptation method (CPAda), we find that the original methods can achieve better inefficiency while maintaining effective coverage, meaning the predicted results are more reliable. Additionally, the error rate has also decreased. The use of three additional metrics (NLL, BS and ECE) for estimating uncertainty further supports that our approach effectively reduces uncertainty and enhances model performance.

## 5.3 ANALYSIS ON CONFORMAL PREDICTION

### 5.3.1 COMPARISONS WITH OTHER NON-EXCHANGEABLE CP METHODS

In Table 4, we compare our TUI with other CP methods including THR (Sadinle et al., 2019), NexCP (Barber et al., 2023) and QTC (Yilmaz & Heckel, 2022). THR is an exchangeable CP method and never considers domain shifts in CTTA, thus it obtains an obvious coverage gap. NexCP and QTC are two non-exchangeable methods, with detailed comparisons available in Sec. 4.3.1. Firstly, for NexCP, we use the same fixed value for domain shift estimation as in the original paper. Since NexCP relies on a fixed value to estimate domain shift, this method is only slightly better than THR and struggles to estimate domain differences in advance during testing. On the other hand, although QTC estimates domain differences in real time, it neglects the unreliability of the current model due to error accumulation over long testing periods. This method yields better results than both THR and NexCP. Next, we compare domain adaptation methods using different CP techniques that similar to the proposed method, and the results show that TUI achieves better accuracy and more precise uncertainty estimates.

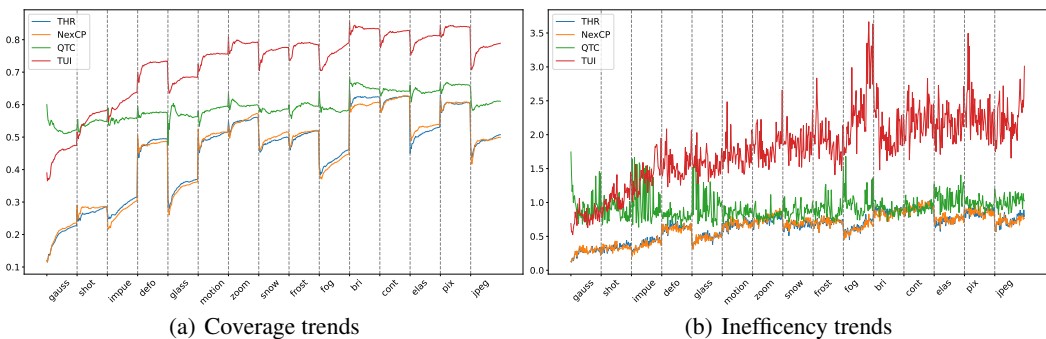

(a) Coverage trends          (b) Inefficiency trends

Figure 2: Visualization of coverage and inefficiency changes on CIFAR100-to-CIFAR100C.

### 5.3.2 CP VISUALIZATION

In Fig. 2, we visualize the coverage and inefficiency of different CP methods. First, as shown in Fig. 2(a), there is a significant disparity in coverage among the different methods, indicating a considerable difference among domains. Existing methods exhibit a substantial coverage gap, particularly THR and NexCP. QTC achieves good coverage during the first domain shift, but as error accumulates, it still struggles to avoid the coverage gap. In comparison, TUI achieves coverage that is similar to QTC in the initial domains, while in subsequent domains, it surpasses QTC in coverage. Second, in Fig. 2(b), we present the trend of inefficiency changes. It is obvious that the comparative methods exhibit good inefficiency despite insufficient coverage, failing to recognize the error accumulation caused by domain shifts, resulting in an overconfidence phenomenon. This indicates that existing methods is difficult to effectively measure uncertainty in ongoing domain change testing scenarios. In contrast, TUI observes error accumulation, with the inefficiency showing an upward trend as the domain changes, indicating that the uncertainty of predictions is continually increasing. After using TUI as guidance for domain adaptation, it is evident that the inefficiency can be effectively reduced, indicating that the overall uncertainty has been controlled.

### 5.3.3 DATA STORAGE ANALYSIS AND COMPARISON WITH REPLAY STRATEGY

As discussed in Sec. 4.3.2, CP-based methods need to maintain an extra calibration set for uncertainty estimation. Although effectively measuring uncertainty is crucial in testing systems, using CP requires a certain amount of memory storage. We analyze the impact of this storage on performance in Table 5 and find that a larger storage capacity leads to better CP performance, as more calibration data provides a more accurate representation of the original data distribution. Additionally, we compare TUI with a classic storage method in continual learning, the source replay strategy, where we use the same samples for replay when conducting adaptation. We find that TUI achieves better accuracy while maintaining the same amount of stored data, which shows the significance of reducing error accumulation in CTTA.

## 6 CONCLUSION

CTTA is prone to error accumulation, where incorrect pseudo-labels can negatively impact subsequent model updates. In this paper, we proposed a simple uncertainty indicator called TUI for the CTTA task based on CP, which generates a set of possible labels for each instance, ensuring that the true label is included within this set with a specified coverage probability. To reduce the coverage gap when domain shifting, we proposed dynamically measuring the domain difference between the target and source domains in continuously changing environments. Moreover, we separate relabeled test pseudo-labels and use them to enhance the adaptation. Experimental results demonstrate that our method effectively estimates the uncertainty for CTTA under a specified coverage probability and improves adaptation performance for various existing CTTA methods. The proposed TUI has the following limitations. First, TUI requires a prepared calibration set for conformal calculation, which may not be satisfied in some situations. Second, TUI performs better than other CP methods in long-term changing domains, but when the domain shifts have an extremely large number it may also lose its effectiveness. In the future, we will further study to solve the two limitations.

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

# A COVERAGE PROOF IN CONFORMAL PREDICTION

In this section, we provide the coverage theorem of conformal prediction.

## A.1 COVERAGE IN EXCHANGEABLE CONFORMAL PREDICTION (WITHOUT DOMAIN SHIFT)

**Theorem 1 (Exchangeable Conformal Prediction (Vovk et al., 2005))** *Assume the calibration set $\mathcal{C}$ and a new data sample $x$ are i.i.d. (or more generally, exchangeable), and the model $\pi$ treats the input data points symmetrically. Given a specified coverage level $\alpha$, the quantile can be calculated by*

$$\tau^* = \text{Quantile}[\mathcal{C}, (1-\alpha)] = \inf\left\{\tau: \frac{1}{|\mathcal{C}|}\sum_{x\in\mathcal{C}}\mathbb{I}_{\{s(\pi(x))<\tau\}} \geq \frac{|\mathcal{C}|+1}{|\mathcal{C}|}(1-\alpha)\right\}. \quad (16)$$

*Then, the conformal prediction set is defined as*

$$\mathcal{P}(x) = \{y|s(\pi(x))<\tau^*\}, \quad (17)$$

*and satisfies*

$$\mathbb{P}(y\in\mathcal{P}(x)) \geq 1-\alpha. \quad (18)$$

**Proof**. The coverage proof of exchangeable CP is following Barber et al. (2023). First, we define the strange data points in the calibration set as an index set:

$$\mathcal{S} = \{i\in[1,n+1]: s(\pi(x_i))>\tau^*\} \quad (19)$$

The strange points are with the largest $\lfloor\alpha(n+1)\rfloor$ non-conformity score. Because of the definition of quantile, it is easy to find that

$$|\mathcal{S}| \leq \alpha(n+1). \quad (20)$$

Then, for a test sample $x_{n+1}$, if it was failed-coverage, say $\hat{y}_{n+1}\notin\mathcal{P}(x_{n+1})$, this means that $s(\pi(x_i))>\tau^*$. Thus, we have the strange probability:

$$p(y_{n+1}\notin\mathcal{P}(x_{n+1})) = p(n+1\in\mathcal{S}) = \mathbb{E}_{i\in[1,n+1]}p(i\in\mathcal{S}) = \frac{|\mathcal{S}|}{n+1} \quad (21)$$

Because of the exchangeability assumation, we have

$$p(y_{n+1}\notin\mathcal{P}(x_{n+1})) \leq \alpha \quad (22)$$

The coverage of exchangeable conformal prediction is obtained proof. $\square$

## A.2 COVERAGE IN NON-EXCHANGEABLE CONFORMAL PREDICTION (WITH DOMAIN SHIFTS)

In this subsection, we prove that why the proposed method can be used to compensate coverage gap in CP when domain shifts. First, following Barber et al. (2023), we give the lower bound of the coverage in non-exchangeable CP when the domain shifts is known.

**Lemma 1 (Coverage gap upper bound)** *Assume that $\forall x\in\mathcal{C}$ and $x^{\text{test}}$ are independent. In a CP approach, the coverage gap can be bounded by the following inequality:*

$$\kappa = (1-\alpha) - \mathbb{P}\{y\in\mathcal{P}(x)\} \leq \frac{2}{n+1}\sum_{i=1}^{n}w_i\cdot\text{d}_{\text{TV}}\left[(x_i,y_i),(x^{\text{test}},y^{\text{test}})\right], \quad (23)$$

*where $\text{d}_{\text{TV}}$ is a total variation distance. $w_i$ is a prespecified importance weight for the $i$-th calibration sample, and is set to 1 in general CP.*

Table 6: Classification error rate (%) for the standard CIFAR10-to-CIFAR10C CTTA task. All results are evaluated with the largest corruption severity level 5 in an online fashion. C1: Gaussian, C2: Shot, C3: Impulse C4: Defocus, C5: Glass, C6: Motion, C7: Zoom, C8: Snow, C9: Frost, C10: Fog, C11: Brightness, C12: Contrast, C13: Elastic, C14: Pixelate, C15: Jpeg. CIFAR100C and ImagenetC use the same setup.

| $\alpha = 0.3$ | C1 | C2 | C3 | C4 | C5 | C6 | C7 | C8 | C9 | C10 | C11 | C12 | C13 | C14 | C15 | Avg |
|---|---|---|---|---|---|---|---|---|---|---|---|---|---|---|---|---|
| Tent | 24.08 | 18.70 | 27.39 | 13.14 | 30.20 | 16.85 | 13.88 | 18.82 | 18.78 | 18.77 | 13.10 | 13.76 | 23.54 | 18.98 | 24.86 | 19.66 |
| + CPAda | 23.84 | 18.61 | 26.85 | 12.76 | 29.50 | 15.44 | 12.40 | 16.24 | 15.28 | 14.48 | 9.36 | 12.19 | 21.31 | 16.42 | 21.81 | 17.96 |
| CoTTA | 23.62 | 20.66 | 28.90 | 11.93 | 27.07 | 13.28 | 11.95 | 16.20 | 15.28 | 14.33 | 9.41 | 12.91 | 18.99 | 14.97 | 18.03 | 17.17 |
| + CPAda | 23.60 | 20.64 | 24.99 | 12.02 | 27.30 | 13.29 | 12.09 | 16.23 | 15.15 | 14.33 | 9.31 | 13.13 | 19.01 | 15.01 | 18.21 | 16.95 |
| SATA | 25.25 | 20.86 | 29.18 | 11.65 | 28.36 | 12.8 | 10.28 | 10.28 | 14.36 | 13.91 | 12.5 | 7.92 | 11.19 | 14.54 | 20.41 | 16.84 |
| + CPAda | 25.06 | 20.51 | 28.33 | 11.51 | 28.15 | 12.76 | 10.18 | 14.30 | 13.84 | 12.34 | 7.80 | 11.04 | 19.20 | 13.79 | 20.36 | 16.61 |
| RMT | 25.20 | 21.08 | 27.92 | 12.69 | 27.81 | 14.93 | 13.14 | 16.78 | 16.47 | 14.95 | 11.26 | 14.22 | 18.26 | 14.65 | 17.52 | 17.79 |
| + CPAda | 24.94 | 20.96 | 27.60 | 12.49 | 27.05 | 14.69 | 12.73 | 16.47 | 16.16 | 14.56 | 11.14 | 13.40 | 18.14 | 14.42 | 17.18 | 17.46 |
| C-CoTTA | 23.39 | 18.27 | 24.15 | 11.89 | 24.65 | 12.39 | 10.00 | 13.35 | 12.78 | 11.82 | 7.70 | 10.51 | 16.89 | 12.08 | 16.55 | 15.09 |
| + CPAda | 23.12 | 18.02 | 23.67 | 11.65 | 25.16 | 12.73 | 10.04 | 13.37 | 12.80 | 11.58 | 7.72 | 9.87 | 16.79 | 11.97 | 16.24 | 14.98 |
| $\alpha = 0.2$ | C1 | C2 | C3 | C4 | C5 | C6 | C7 | C8 | C9 | C10 | C11 | C12 | C13 | C14 | C15 | Avg |
| Tent | 24.08 | 18.70 | 27.39 | 13.14 | 30.20 | 16.85 | 13.88 | 18.82 | 18.78 | 18.77 | 13.10 | 13.76 | 23.54 | 18.98 | 24.86 | 19.66 |
| + CPAda | 24.40 | 19.03 | 27.91 | 13.21 | 29.95 | 15.47 | 12.70 | 17.77 | 16.52 | 15.62 | 9.70 | 12.48 | 21.45 | 16.48 | 21.23 | 18.26 |
| CoTTA | 23.62 | 20.66 | 28.90 | 11.93 | 27.07 | 13.28 | 11.95 | 16.20 | 15.28 | 14.33 | 9.41 | 12.91 | 18.99 | 14.97 | 18.03 | 17.17 |
| + CPAda | 23.52 | 20.90 | 25.62 | 12.32 | 27.32 | 13.28 | 12.06 | 16.17 | 15.57 | 14.35 | 9.93 | 13.70 | 18.75 | 14.13 | 17.57 | 17.01 |
| SATA | 25.25 | 20.86 | 29.18 | 11.65 | 28.36 | 12.8 | 10.28 | 10.28 | 14.36 | 13.91 | 12.5 | 7.92 | 11.19 | 14.54 | 20.41 | 16.84 |
| + CPAda | 25.00 | 20.34 | 28.24 | 11.50 | 28.20 | 12.60 | 10.11 | 14.28 | 13.65 | 12.28 | 7.69 | 10.82 | 19.30 | 14.00 | 20.30 | 16.55 |
| RMT | 25.20 | 21.08 | 27.92 | 12.69 | 27.81 | 14.93 | 13.14 | 16.78 | 16.47 | 14.95 | 11.26 | 14.22 | 18.26 | 14.65 | 17.52 | 17.79 |
| + CPAda | 25.09 | 21.00 | 27.66 | 12.42 | 27.36 | 14.55 | 12.93 | 16.60 | 15.90 | 14.62 | 11.18 | 13.90 | 18.31 | 14.42 | 17.06 | 17.53 |
| C-CoTTA | 23.39 | 18.27 | 24.15 | 11.89 | 24.65 | 12.39 | 10.00 | 13.35 | 12.78 | 11.82 | 7.70 | 10.51 | 16.89 | 12.08 | 16.55 | 15.09 |
| + CPAda | 22.99 | 17.97 | 23.48 | 11.67 | 24.48 | 12.47 | 9.93 | 13.34 | 12.43 | 11.62 | 7.77 | 10.19 | 16.52 | 12.11 | 16.18 | 14.88 |
| $\alpha = 0.1$ | C1 | C2 | C3 | C4 | C5 | C6 | C7 | C8 | C9 | C10 | C11 | C12 | C13 | C14 | C15 | Avg |
| Tent | 24.08 | 18.70 | 27.39 | 13.14 | 30.20 | 16.85 | 13.88 | 18.82 | 18.78 | 18.77 | 13.10 | 13.76 | 23.54 | 18.98 | 24.86 | 19.66 |
| + CPAda | 24.42 | 18.82 | 27.58 | 12.85 | 29.56 | 14.04 | 11.38 | 15.96 | 15.70 | 14.53 | 8.99 | 11.77 | 20.70 | 15.69 | 22.00 | 17.60 |
| CoTTA | 23.62 | 20.66 | 28.90 | 11.93 | 27.07 | 13.28 | 11.95 | 16.20 | 15.28 | 14.33 | 9.41 | 12.91 | 18.99 | 14.97 | 18.03 | 17.17 |
| + CPAda | 23.33 | 20.47 | 25.19 | 11.96 | 26.72 | 13.05 | 11.62 | 15.80 | 14.85 | 13.92 | 9.73 | 12.52 | 18.11 | 13.73 | 16.95 | 16.53 |
| SATA | 25.25 | 20.86 | 29.18 | 11.65 | 28.36 | 12.8 | 10.28 | 10.28 | 14.36 | 13.91 | 12.5 | 7.92 | 11.19 | 14.54 | 20.41 | 16.84 |
| + CPAda | 24.97 | 20.33 | 28.45 | 11.46 | 28.16 | 12.64 | 10.00 | 14.41 | 13.65 | 12.16 | 7.66 | 10.68 | 19.24 | 14.06 | 20.11 | 16.53 |
| RMT | 25.20 | 21.08 | 27.92 | 12.69 | 27.81 | 14.93 | 13.14 | 16.78 | 16.47 | 14.95 | 11.26 | 14.22 | 18.26 | 14.65 | 17.52 | 17.79 |
| + CPAda | 24.84 | 21.23 | 27.84 | 12.80 | 27.70 | 14.88 | 13.05 | 16.75 | 16.04 | 15.31 | 11.47 | 14.36 | 18.41 | 14.75 | 16.95 | 17.76 |
| C-CoTTA | 23.39 | 18.27 | 24.15 | 11.89 | 24.65 | 12.39 | 10.00 | 13.35 | 12.78 | 11.82 | 7.70 | 10.51 | 16.89 | 12.08 | 16.55 | 15.09 |
| + CPAda | 23.12 | 18.02 | 23.67 | 11.65 | 25.16 | 12.73 | 10.04 | 13.37 | 12.80 | 11.58 | 7.72 | 9.87 | 16.79 | 11.97 | 16.24 | 14.98 |

**Proof**. Let $\mathcal{X} = \mathcal{C} \cup \{(x^{\text{test}}, y^{\text{test}})\}$. Because $\forall x \in \mathcal{C}$ and $x^{\text{test}}$ are independent, we have

$$\kappa = (1 - \alpha) - \mathbb{P}\{y \in \mathcal{P}(x)\}$$

$$\leq \frac{1}{n+1} \sum_{i=1}^{n+1} w_i \cdot \mathrm{d}_{\mathrm{TV}}\left(\mathcal{X}, (x_1, y_1)\right)$$

$$\leq \frac{1}{n+1} \sum_{i=1}^{n} w_i \cdot \left(2\mathrm{d}_{\mathrm{TV}}\left[(x_i, y_i), (x^{\text{test}}, y^{\text{test}})\right] - \mathrm{d}_{\mathrm{TV}}\left[(x_i, y_i), (x^{\text{test}}, y^{\text{test}})\right]^2\right) \quad (24)$$

$$\leq \frac{2}{n+1} \sum_{i=1}^{n} w_i \cdot \mathrm{d}_{\mathrm{TV}}\left[(x_i, y_i), (x^{\text{test}}, y^{\text{test}})\right],$$

where the second inequality can be obtained by the maximal coupling theorem (Den Hollander, 2012). That is, for two independent random variables $x$ and $y$, if we have another two independent random variables $\hat{x}$ and $\hat{y}$ and $(\hat{x}, \hat{y})$ is a maximal coupling for $(x, y)$, then we have $d_{\mathrm{TV}}(x, y) = p(\hat{x} \neq \hat{y})$.

**Theorem 2 (Exchangeable Conformal Prediction with Known Shifts (Barber et al., 2023))**
*Assume the calibration set $\mathcal{C}$ is i.i.d., but a new data sample $x$ is drawn from a different distribution. Given a specified coverage level $\alpha$, the quantile can be calculated by*

$$\tau^* = \mathrm{Quantile}[\mathcal{C}, (1 - \alpha)] = \inf\left\{\tau : \frac{1}{|\mathcal{C}|} \sum_{x \in \mathcal{C}} \mathbb{I}_{\{s(\pi(x)) < \tau\}} \geq \frac{|\mathcal{C}| + 1}{|\mathcal{C}|}(1 - \alpha)\right\}. \quad (25)$$

*Then, the conformal prediction set is defined as*

$$\mathcal{P}(x) = \{y | s(\pi(x)) < \tau^*\}, \quad (26)$$

Table 7: Classification error rate (%) for the standard CIFAR100-to-CIFAR100C CTTA task. All results are evaluated with the largest corruption severity level 5 in an online fashion. C1: Gaussian, C2: Shot, C3: Impulse C4: Defocus, C5: Glass, C6: Motion, C7: Zoom, C8: Snow, C9: Frost, C10: Fog, C11: Brightness, C12: Contrast, C13: Elastic, C14: Pixelate, C15: Jpeg.

| $\alpha = 0.3$ | C1 | C2 | C3 | C4 | C5 | C6 | C7 | C8 | C9 | C10 | C11 | C12 | C13 | C14 | C15 | Avg |
|---|---|---|---|---|---|---|---|---|---|---|---|---|---|---|---|---|
| Tent | 37.3 | 35.69 | 41.97 | 38.33 | 50.64 | 46.6 | 47.58 | 57.84 | 63.72 | 71.59 | 74.17 | 86.3 | 91.69 | 93.29 | 95.25 | 62.13 |
| + CPAda | 36.49 | 33.02 | 36.2 | 28.89 | 39.85 | 33.43 | 32.19 | 39.59 | 42.95 | 54.02 | 57.21 | 70.27 | 77.21 | 77.71 | 84.68 | 49.58 |
| CoTTA | 40.13 | 37.08 | 39.26 | 26.87 | 36.77 | 28.08 | 26.36 | 32.38 | 31.99 | 39.84 | 25.53 | 26.97 | 31.71 | 27.92 | 32.52 | 32.23 |
| + CPAda | 39.64 | 36.96 | 38.77 | 26.85 | 36.08 | 28.23 | 26.45 | 32.29 | 31.39 | 38.5 | 25.61 | 27.49 | 30.89 | 27.74 | 31.79 | 31.91 |
| SATA | 38.27 | 35.76 | 38.23 | 27.12 | 37.13 | 28.68 | 25.86 | 31.01 | 31.03 | 35.13 | 24.11 | 26.53 | 32.09 | 28.94 | 36.09 | 31.73 |
| + CPAda | 36.72 | 34.42 | 36.24 | 26.28 | 36.04 | 28.14 | 25.39 | 29.67 | 30.00 | 33.36 | 23.54 | 25.75 | 31.62 | 28.16 | 35.23 | 30.70 |
| RMT | 39.52 | 36.49 | 37.33 | 26.70 | 35.10 | 28.86 | 26.88 | 29.99 | 30.16 | 33.00 | 26.87 | 28.96 | 29.76 | 28.40 | 32.15 | 31.34 |
| + CPAda | 39.58 | 36.74 | 37.42 | 26.94 | 34.84 | 28.56 | 26.86 | 30.20 | 29.94 | 32.71 | 26.89 | 28.70 | 29.50 | 28.38 | 30.76 | 31.20 |
| C-CoTTA | 38.11 | 35.21 | 36.30 | 27.50 | 35.06 | 28.45 | 25.83 | 29.07 | 29.06 | 31.34 | 24.35 | 26.52 | 28.46 | 26.37 | 31.83 | 30.23 |
| + CPAda | 37.54 | 34.11 | 35.18 | 27.94 | 34.11 | 28.71 | 26.36 | 28.61 | 28.45 | 30.71 | 25.20 | 26.40 | 28.03 | 26.28 | 30.57 | 29.88 |
| $\alpha = 0.2$ | C1 | C2 | C3 | C4 | C5 | C6 | C7 | C8 | C9 | C10 | C11 | C12 | C13 | C14 | C15 | Avg |
| Tent | 37.30 | 35.69 | 41.97 | 38.33 | 50.64 | 46.60 | 47.58 | 57.84 | 63.72 | 71.59 | 74.17 | 86.30 | 91.69 | 93.29 | 95.25 | 62.13 |
| + CPAda | 36.52 | 32.93 | 36.21 | 28.75 | 40.59 | 34.75 | 34.26 | 43.32 | 49.43 | 61.65 | 63.26 | 73.68 | 81.65 | 83.28 | 89.63 | 52.66 |
| CoTTA | 40.13 | 37.08 | 39.26 | 26.87 | 36.77 | 28.08 | 26.36 | 32.38 | 31.99 | 39.84 | 25.53 | 26.97 | 31.71 | 27.92 | 32.52 | 32.23 |
| + CPAda | 39.84 | 36.99 | 38.41 | 26.85 | 36.39 | 28.21 | 26.49 | 31.98 | 31.64 | 38.56 | 25.95 | 27.50 | 31.69 | 28.15 | 32.05 | 32.05 |
| SATA | 38.27 | 35.76 | 38.23 | 27.12 | 37.13 | 28.68 | 25.86 | 31.01 | 31.03 | 35.13 | 24.11 | 26.53 | 32.09 | 28.94 | 36.09 | 31.73 |
| + CPAda | 36.61 | 33.71 | 35.66 | 26.1 | 36.26 | 28.05 | 25.16 | 29.28 | 29.99 | 33.54 | 23.42 | 25.67 | 31.13 | 27.87 | 34.76 | 30.48 |
| RMT | 39.52 | 36.49 | 37.33 | 26.70 | 35.10 | 28.86 | 26.88 | 29.99 | 30.16 | 33.00 | 26.87 | 28.96 | 29.76 | 28.40 | 32.15 | 31.34 |
| + CPAda | 39.74 | 36.49 | 37.33 | 26.75 | 35.18 | 28.83 | 27.24 | 30.31 | 29.97 | 32.94 | 27.12 | 28.80 | 29.79 | 28.46 | 31.13 | 31.34 |
| C-CoTTA | 38.11 | 35.21 | 36.30 | 27.50 | 35.06 | 28.45 | 25.83 | 29.07 | 29.06 | 31.34 | 24.35 | 26.52 | 28.46 | 26.37 | 31.83 | 30.23 |
| + CPAda | 37.37 | 33.78 | 35.34 | 28.12 | 33.20 | 28.40 | 26.28 | 27.90 | 27.94 | 30.08 | 24.98 | 26.39 | 27.35 | 25.85 | 29.85 | 29.52 |
| $\alpha = 0.1$ | C1 | C2 | C3 | C4 | C5 | C6 | C7 | C8 | C9 | C10 | C11 | C12 | C13 | C14 | C15 | Avg |
| Tent | 37.30 | 35.69 | 41.97 | 38.33 | 50.64 | 46.6 | 47.58 | 57.84 | 63.72 | 71.59 | 74.17 | 86.30 | 91.69 | 93.29 | 95.25 | 62.13 |
| + CPAda | 36.25 | 33.47 | 38.39 | 32.49 | 46.42 | 42.88 | 45.22 | 56.06 | 60.56 | 67.70 | 64.85 | 72.49 | 79.75 | 80.65 | 86.02 | 56.21 |
| CoTTA | 40.13 | 37.08 | 39.26 | 26.87 | 36.77 | 28.08 | 26.36 | 32.38 | 31.99 | 39.84 | 25.53 | 26.97 | 31.71 | 27.92 | 32.52 | 32.23 |
| + CPAda | 39.84 | 36.99 | 38.41 | 26.85 | 36.39 | 28.21 | 26.49 | 31.98 | 31.64 | 38.56 | 25.95 | 27.50 | 31.69 | 28.15 | 32.05 | 32.05 |
| SATA | 38.27 | 35.76 | 38.23 | 27.12 | 37.13 | 28.68 | 25.86 | 31.01 | 31.03 | 35.13 | 24.11 | 26.53 | 32.09 | 28.94 | 36.09 | 31.73 |
| + CPAda | 36.01 | 33.44 | 35.35 | 26.08 | 35.81 | 27.84 | 24.95 | 29.57 | 29.63 | 33.81 | 23.38 | 25.23 | 30.99 | 27.48 | 34.32 | 30.26 |
| RMT | 39.52 | 36.49 | 37.33 | 26.70 | 35.10 | 28.86 | 26.88 | 29.99 | 30.16 | 33.00 | 26.87 | 28.96 | 29.76 | 28.40 | 32.15 | 31.34 |
| + CPAda | 39.74 | 36.41 | 37.14 | 27.25 | 35.11 | 28.55 | 27.01 | 30.03 | 29.62 | 32.57 | 26.54 | 28.71 | 29.46 | 28.14 | 30.83 | 31.14 |
| C-CoTTA | 38.11 | 35.21 | 36.30 | 27.50 | 35.06 | 28.45 | 25.83 | 29.07 | 29.06 | 31.34 | 24.35 | 26.52 | 28.46 | 26.37 | 31.83 | 30.23 |
| + CPAda | 36.88 | 33.57 | 34.60 | 27.04 | 32.98 | 27.62 | 25.23 | 27.72 | 27.85 | 30.45 | 24.15 | 25.96 | 27.56 | 25.72 | 31.40 | 29.25 |

*and satisfies a coverage lower bound:*

$$\mathbb{P}(y \in \mathcal{P}(x)) \geq 1 - \alpha - \frac{2}{n} \sum_{i=1}^{n} w_i \cdot \mathrm{d}_{\mathrm{TV}} \left[ (x_i, y_i), (x^{\text{test}}, y^{\text{test}}) \right]. \tag{27}$$

**Proof**. This theorem can be easily obtained from Lemma 1.

### A.3 COVERAGE OF TUI WITH DOMAIN SHIFTS

However, Theorem 2 is only appropriate for known domain difference. When the domain differences are unknown in test time, it is difficult to obtain a certain coverage lower bound. This explains why NexCP performs poorly in the CTTA task. QTC has designed a dynamic method for estimating domain differences, making it more suitable for testing compared to NexCP. However, the CTTA task requires multiple domain changes, which significantly impacts the model's ability to estimate domain differences due to error accumulation. Specifically, we compute the joint distribution difference of current data and calibration data between both the source and current model.

In TUI, we dynamic evaluate the domain difference between the source data and the current test data. To mitigate the effect of error accumulation, we consider both model and data difference. We use the Jensen-Shannon (JS) divergence as the metric. Joint feature representation captures correlations between different features, providing a more holistic view of the data distribution and how different models process it. The joint distribution can better reflect subtle differences between domains, enhancing the precision of JS divergence measures. Moreover, comparing joint feature distributions allows for a more detailed assessment of how much the current model has gained compared to the source model.

Table 8: Classification error rate (%) for the standard ImageNet-to-ImageNetC CTTA task. All results are evaluated with the largest corruption severity level 5 in an online fashion. C1: Gaussian, C2: Shot, C3: Impulse C4: Defocus, C5: Glass, C6: Motion, C7: Zoom, C8: Snow, C9: Frost, C10: Fog, C11: Brightness, C12: Contrast, C13: Elastic, C14: Pixelate, C15: Jpeg.

| $\alpha = 0.3$ | C1 | C2 | C3 | C4 | C5 | C6 | C7 | C8 | C9 | C10 | C11 | C12 | C13 | C14 | C15 | Avg |
|---|---|---|---|---|---|---|---|---|---|---|---|---|---|---|---|---|
| Tent | 81.32 | 74.76 | 72.96 | 77.20 | 74.14 | 66.24 | 55.70 | 61.66 | 63.08 | 51.44 | 38.16 | 71.82 | 51.06 | 47.56 | 53.22 | 62.69 |
| + CPAda | 81.26 | 74.66 | 72.62 | 77.20 | 73.90 | 65.98 | 55.68 | 61.54 | 63.02 | 51.22 | 38.28 | 70.96 | 50.78 | 47.62 | 52.80 | 62.50 |
| CoTTA | 85.12 | 82.62 | 81.52 | 83.32 | 81.30 | 72.00 | 63.76 | 63.76 | 64.28 | 52.50 | 41.54 | 70.86 | 51.16 | 45.20 | 49.30 | 65.88 |
| + CPAda | 84.88 | 82.26 | 81.06 | 82.80 | 80.16 | 71.00 | 61.54 | 63.50 | 63.38 | 52.24 | 41.44 | 69.46 | 51.10 | 45.82 | 49.62 | 65.35 |
| SATA | 80.72 | 79.20 | 77.90 | 79.14 | 78.14 | 67.28 | 56.02 | 58.02 | 64.34 | 47.18 | 34.38 | 73.00 | 51.36 | 45.74 | 51.84 | 62.95 |
| + CPAda | 79.40 | 78.46 | 77.76 | 79.06 | 77.74 | 65.98 | 56.10 | 58.42 | 63.82 | 46.38 | 34.28 | 72.00 | 50.96 | 44.92 | 51.42 | 62.45 |
| RMT | 80.06 | 76.42 | 73.18 | 75.80 | 73.06 | 64.94 | 57.22 | 56.20 | 58.74 | 48.76 | 40.50 | 59.32 | 47.20 | 43.70 | 48.12 | 60.21 |
| + CPAda | 81.18 | 76.62 | 74.22 | 76.14 | 73.62 | 63.82 | 56.08 | 56.60 | 57.90 | 48.56 | 38.92 | 58.62 | 47.24 | 43.48 | 45.52 | 59.90 |
| C-CoTTA | 76.70 | 74.24 | 71.90 | 76.44 | 73.86 | 66.22 | 57.70 | 55.92 | 60.96 | 49.36 | 39.42 | 63.24 | 49.46 | 43.04 | 45.08 | 60.24 |
| + CPAda | 76.88 | 73.04 | 69.92 | 75.20 | 72.50 | 65.58 | 57.36 | 55.28 | 59.76 | 49.72 | 40.76 | 62.58 | 49.18 | 44.04 | 44.42 | 59.75 |

| $\alpha = 0.2$ | C1 | C2 | C3 | C4 | C5 | C6 | C7 | C8 | C9 | C10 | C11 | C12 | C13 | C14 | C15 | Avg |
|---|---|---|---|---|---|---|---|---|---|---|---|---|---|---|---|---|
| Tent | 81.32 | 74.76 | 72.96 | 77.20 | 74.14 | 66.24 | 55.70 | 61.66 | 63.08 | 51.44 | 38.16 | 71.82 | 51.06 | 47.56 | 53.22 | 62.69 |
| + CPAda | 81.14 | 74.54 | 72.64 | 77.02 | 73.88 | 65.96 | 55.88 | 61.70 | 63.08 | 51.36 | 38.30 | 71.24 | 50.86 | 47.54 | 52.76 | 62.53 |
| CoTTA | 85.12 | 82.62 | 81.52 | 83.32 | 81.30 | 72.00 | 63.76 | 63.76 | 64.28 | 52.50 | 41.54 | 70.86 | 51.16 | 45.20 | 49.30 | 65.88 |
| + CPAda | 84.40 | 82.38 | 80.90 | 82.50 | 82.50 | 70.56 | 61.42 | 63.70 | 64.18 | 51.68 | 40.40 | 67.40 | 52.04 | 45.56 | 48.92 | 65.24 |
| SATA | 80.72 | 79.20 | 77.90 | 79.14 | 78.14 | 67.28 | 56.02 | 58.02 | 64.34 | 47.18 | 34.38 | 73.00 | 51.36 | 45.74 | 51.84 | 62.95 |
| + CPAda | 81.00 | 79.28 | 77.86 | 79.38 | 78.22 | 66.80 | 56.52 | 58.52 | 63.88 | 47.18 | 34.52 | 73.10 | 51.68 | 45.14 | 52.38 | 63.03 |
| RMT | 80.06 | 76.42 | 73.18 | 75.80 | 73.06 | 64.94 | 57.22 | 56.20 | 58.74 | 48.76 | 40.50 | 59.32 | 47.20 | 43.70 | 48.12 | 60.21 |
| + CPAda | 80.14 | 75.98 | 73.52 | 75.64 | 73.20 | 63.94 | 56.98 | 56.70 | 58.54 | 48.80 | 40.14 | 58.46 | 47.04 | 43.72 | 45.76 | 59.90 |
| C-CoTTA | 76.70 | 74.24 | 71.90 | 76.44 | 73.86 | 66.22 | 57.70 | 55.92 | 60.96 | 49.36 | 39.42 | 63.24 | 49.46 | 43.04 | 45.08 | 60.24 |
| + CPAda | 76.08 | 73.24 | 70.32 | 75.46 | 73.70 | 66.26 | 58.32 | 55.84 | 59.84 | 49.64 | 40.32 | 63.18 | 49.76 | 44.58 | 41.44 | 59.87 |

| $\alpha = 0.1$ | C1 | C2 | C3 | C4 | C5 | C6 | C7 | C8 | C9 | C10 | C11 | C12 | C13 | C14 | C15 | Avg |
|---|---|---|---|---|---|---|---|---|---|---|---|---|---|---|---|---|
| Tent | 81.32 | 74.76 | 72.96 | 77.20 | 74.14 | 66.24 | 55.70 | 61.66 | 63.08 | 51.44 | 38.16 | 71.82 | 51.06 | 47.56 | 53.22 | 62.69 |
| + CPAda | 81.22 | 74.76 | 72.94 | 77.12 | 74.02 | 66.20 | 55.70 | 61.66 | 63.00 | 51.42 | 38.20 | 71.40 | 50.94 | 47.50 | 52.92 | 62.60 |
| CoTTA | 85.12 | 82.62 | 81.52 | 83.32 | 81.30 | 72.00 | 63.76 | 63.76 | 64.28 | 52.50 | 41.54 | 70.86 | 51.16 | 45.20 | 49.30 | 65.88 |
| + CPAda | 85.00 | 82.52 | 81.30 | 82.54 | 80.58 | 71.26 | 61.82 | 63.16 | 63.80 | 52.32 | 41.34 | 69.18 | 51.12 | 45.58 | 49.10 | 65.37 |
| SATA | 80.72 | 79.20 | 77.90 | 79.14 | 78.14 | 67.28 | 56.02 | 58.02 | 64.34 | 47.18 | 34.38 | 73.00 | 51.36 | 45.74 | 51.84 | 62.95 |
| + CPAda | 81.60 | 78.82 | 76.48 | 78.04 | 76.36 | 65.72 | 56.22 | 57.64 | 62.54 | 46.60 | 35.18 | 70.94 | 51.28 | 45.00 | 51.48 | 62.26 |
| RMT | 80.06 | 76.42 | 73.18 | 75.80 | 73.06 | 64.94 | 57.22 | 56.20 | 58.74 | 48.76 | 40.50 | 59.32 | 47.20 | 43.70 | 48.12 | 60.21 |
| + CPAda | 81.60 | 77.44 | 74.28 | 76.30 | 73.14 | 63.48 | 55.56 | 56.18 | 58.04 | 48.82 | 39.08 | 58.68 | 47.06 | 43.78 | 45.34 | 59.92 |
| C-CoTTA | 76.70 | 74.24 | 71.90 | 76.44 | 73.86 | 66.22 | 57.70 | 55.92 | 60.96 | 49.36 | 39.42 | 63.24 | 49.46 | 43.04 | 45.08 | 60.24 |
| + CPAda | 76.00 | 73.54 | 69.72 | 76.06 | 73.36 | 65.44 | 57.10 | 55.00 | 59.66 | 50.16 | 39.96 | 62.62 | 48.88 | 43.70 | 44.12 | 59.69 |

## B DETAILED RESULTS

In our experiments, we employ the CIFAR10C, CIFAR100C, and ImageNetC datasets as benchmarks to assess the robustness of classification models. Each dataset comprises 15 distinct types of corruption, each applied at five different levels of severity (from 1 to 5). These corruptions are systematically applied to test images from the original CIFAR10 and CIFAR100 datasets, as well as validation images from the original ImageNet dataset. The 15 types of corruption are Gaussian, Shot, Impulse, Defocus, Glass, Motion, Zoom, Snow, Frost, Fog, Brightness, Contrast, Elastic, Pixelate, Jpeg. We show the detailed error results for each type of corruption in Tables 6, 7 and 8.

