# OpenReview forum: "TUI: A Conformal Uncertainty Indicator for Continual Test-Time Adaptation"
_ICLR.cc/2025/Conference — ICLR 2025 Conference Withdrawn Submission_

### Official Review · Reviewer_qpy6 · 2024-10-30

**Soundness:** 2
**Presentation:** 2
**Contribution:** 2
**Rating:** 3
**Confidence:** 4

**Summary:**

The paper introduces a novel method, TUI, for estimating uncertainty in Continual Test-Time Adaptation (CTTA) using Conformal Prediction (CP). This approach aims to enhance model adaptation by filtering unreliable pseudo-labels through uncertainty estimation, which is crucial for applications where model predictions must be robust across domain shifts.

**Strengths:**

1. The application of CP to estimate uncertainty in CTTA is new. Unlike traditional methods that may not handle domain shifts effectively, TUI provides a model-agnostic framework that enhances prediction reliability by including uncertainty measures.
2.  Evaluating both the accuracy and the uncertainty of predictions provides a more comprehensive assessment of model performance in test-time adaptation scenarios. This dual approach is crucial for understanding the practical utility of TUI in real-world applications.

**Weaknesses:**

1. (**major**)The experimental comparison includes only a few methods like Tent, CoTTA, SATA, RMT, and C-CoTTA. Recent advancements in CTTA such as [1-3] in recent publications are not considered. This omission may raise concerns about whether TUI can maintain its performance advantages when compared to the latest methods.
2. (**minor**) TUI's requirement for labeled source data for its calibration set may limit its applicability in real-world scenarios where such data is unavailable or impractical to use such as CLIP. (**major**) This limitation also complicates direct comparisons with other CTTA methods that do not use source domain knowledge, potentially skewing the fairness of these comparisons .
3. (**major**)The paper claims that the TUI-guided adaptation method (CPAda) improves accuracy and reduces uncertainty. However, the evidence provided across three datasets and various settings (different alpha values) does not consistently support these claims. Notably, clear improvement is observed only when CPAda is applied to Tent on CIFAR100. In contrast, other applications show minimal or no benefits—for instance, when CPAda is applied to CoTTA and C-CoTTA on CIFAR100, as well as CoTTA, SATA, and C-CoTTA on CIFAR10, not only do the accuracy improvements lack significance, but all three uncertainty metrics (NLL, BS, ECE) actually increase.
4. (**minor**) The paper does not provide an analysis of the computational efficiency of the TUI-guided method (CPAda), which is crucial given the computational demands of TTA methods. For each incoming batch, TUI requires that the forward process be conducted on the new batch and calibration dataset with both the original source model and adapted model. This quadruples the computational load compared to simpler methods like CoTTA, which does not require multiple forward passes per batch. This lack of information makes it difficult to assess the practicality of deploying TUI in environments where resources are constrained or efficiency is a critical factor.

[1] ViDA: Homeostatic Visual Domain Adapter for Continual Test Time Adaptation, ICLR 2024

[2] Becotta: Input-dependent online blending of experts for continual test-time adaptation, ICML 2024

[3] Decorate the Newcomers:Visual Domain Prompt for Continual Test Time Adaptation, AAAI 2023

**Questions:**

1. A grammar error in line 310.
2. What are the deinitions of $s(y|\pi(x))$ in formula (9) and $C(x)$ in formula (14)?
3. Can you clarify the design behind formula (4), where the dimensionality doubles the number of labels? How to interpret the derived probablities? Could the order of operations be altered, for instance, concatenating after applying the softmax?
4. The sentence from lines 461-464 is confusing, as it mentions good coverage paired with high inefficiency in the first point, contradicting the claim of excellent inefficiency but low coverage in the second point.
5. Could you provide more details on how to interpret the metrics COV and INE? How to interpret these two metrics? How are these metrics related to the performance and uncertainty assessments in the CTTA model?

---

> ### Author Response · Authors · 2024-11-25
>
> # Responses to the Weaknesses
> ## A1
>
>
> - The objective of this paper is not to compete with the latest SOTA CTTA methods but to propose a **plug-and-play approach** for measuring the uncertainty of test results in existing methods, primarily assessing whether it can effectively address the coverage gap in conformal prediction. As a plug-and-play method, our approach has been constructed on top of some existing CTTA methods (Table 1,2,3), and the results demonstrate its effectiveness.
> - Our method designs a conformal prediction (CP) technique, and we **have compared it with SOTA CP methods** including NexCP and QTC. The comparative results are shown in Table 4.
> - To address the reviewer's concerns, we use ViDA to evaluate the uncertainty measurement outcomes, and the results are as follows:
> | Method     | ERR   | COV   | INE  | NLL  | BS   | ECE   |
> |------------|-------|-------|------|------|------|-------|
> | α = 0.3        |       |       |      |      |      |       |
> | vida+TUI       | 18.64 | 44.11 | 0.48 | 0.94 | 0.14 | 0.14  |
> | vida+TUI+CPAda | 18.61 | 69.49 | 2.72 | 0.95 | 0.15 | 0.16  |
> | α = 0.2        |       |       |      |      |      |       |
> | vida+TUI       | 18.64 | 54.29 | 0.65 | 0.94 | 0.14 | 0.14  |
> | vida+TUI+CPAda | 18.63 | 76.23 | 2.86 | 0.95 | 0.15 | 0.17  |
> | α = 0.1        |       |       |      |      |      |       |
> | vida+TUI       | 18.64 | 67.45 | 1    | 0.94 | 0.14 | 0.14  |
> | vida+TUI+CPAda | 18.53 | 85.29 | 3.27 | 0.95 | 0.15 | 0.17  |
>
> ## A2
>
> Our paper is plug-and-play, and builds on existing CTTA methods, aiming to assess the credibility of upcoming test results with just a small set of calibration data and to improve the adaptation effect using this assessment. The calibration set can be derived from the training set or constructed separately, and is entirely independent of the model itself.
> - **Fairness**: We have considered that previous methods used all training data during pre-training. Therefore, all comparative methods (as shown in all experimental charts and tables) in our paper employ calibration sets constructed in the same manner for conformal prediction, ensuring there is no issue of unfairness.
> - **Privacy**: The construction of the calibration set can be independent of the training set, serving merely as an additional reference to assist in measuring the uncertainty of test results. To illustrate this, we separate the source training data into training and calibration datasets during the pre-training phase. We train the pre-trained model only on the separated training data, without training on the calibration data, thus eliminating privacy concerns. The results are as follows:
>
> | With seperated calibration set          | ERR   | COV   | INE  | NLL  | BS   | ECE   |
> |-----------------|-------|-------|------|------|------|-------|
> | tent+TUI        | 19.01 | 70.68 | 0.84 | 1.67 | 0.16 | 0.16  |
> | tent+TUI+CPAda  | 18.89 | 77.76 | 1.06 | 1.75 | 0.17 | 0.19  |
> | CoTTA+TUI       | 17.44 | 83.83 | 1.68 | 0.57 | 0.11 | 0.05  |
> | CoTTA+TUI+CPAda | 16.61 | 78.71 | 1.11 | 0.64 | 0.12 | 0.12  |
> | SATA+TUI        | 16.65 | 47.46 | 0.51 | 0.57 | 0.1  | 0.03  |
> | SATA+TUI+CPAda  | 16.51 | 77.6  | 0.98 | 0.67 | 0.11 | 0.1   |
> | RMT+TUI         | 17.36 | 88.35 | 1.88 | 0.69 | 0.12 | 0.05  |
> | RMT+TUI+CPAda   | 16.7  | 78.56 | 0.98 | 0.75 | 0.12 | 0.14  |
>
> ## A3
>
> It is worth noting that the primary contribution of this paper is the proposal of a plug-and-play method for measuring the uncertainty of test results in existing CTTA methods. The main evaluation lies in effectively addressing the coverage gap in CP (as shown in the COV column) and achieving coverage close to the expected α.
>
> Uncertainty estimation is only credible if the coverage gap (COV column) is addressed, which is the major contribution of this work. In the examples mentioned by the reviewer, including CoTTA and C-CoTTA on CIFAR100, as well as CoTTA, SATA, and C-CoTTA on CIFAR10, the CPAda method shows significant improvements in COV, making the uncertainty estimation more reliable.
>
>
> ## A4
>
> The computational efficiency of CPAda is equivalent to that of standard adaptation methods. It only adds sample-level weights during backpropagation (as shown in Eq. 11), and the complexity of calculating these weights is negligible.
>
> However, the proposed TUI does increase computational time since it requires additional forward passes on the calibration set. In practice, the added time is minimal. As our method focuses on estimating the uncertainty of test results—something not addressed in previous CTTA works—we believe the added time is acceptable. **The benefits of timely risk mitigation far outweigh the slight increase in computational cost**.
>
> | alpha=0.2      | cotta  | rmt    | ccotta | ViDA    |
> |----------------|--------|--------|--------|---------|
> | #           | 1.0695 | 1.0307 | 1.1262 | 1.2542  |
> | #+TUI       | 1.2097 | 1.1643 | 1.2602 | 1.2599  |
> | #+TUI+CPAda | 1.2118 | 1.2155 | 1.2807 | 1.2904  |

---

> ### Author Response · Authors · 2024-11-25
>
> # Answers to the Questions
> ## A1
> Thanks for your suggestion.
>
> ## A2
> $ s(y|\pi(x)) $, referred to in the paper as the **non-conformity score**, specifically represents the score for each category $ y $ as defined in Eq. 7. Eq. 9 indicates the process of collecting all categories with scores below the given threshold and including them in the prediction set.
>
> ## A3
> The purpose of the Concat operation is to simultaneously consider the outputs of both the source model and the current model for the same test sample $ x $. The use of softmax aims to project the concatenated results into the probability space, ensuring that each value in the output comes from a valid probability distribution.  Eq. 4 is the core design of the proposed method, as it accounts for **model shift** when constructing conformal prediction (CP). Combined with Eq. 5, the method jointly considers **model shift** and **data shift**, thereby making the CP results more reliable.
>
> ## A4 & A5
> These two issues are highly related and will be addressed together.
>
> - As shown in Eq. 14, **coverage (COV)** and **inefficiency (INE)** are the two most critical metrics in conformal prediction. COV represents the actual coverage, indicating the proportion of true labels included in the prediction set. Ideally, COV should correspond to the acceptable error rate $\alpha$ provided by the user. For example, if the user specifies $\alpha = 0.2$, the closer COV is to 80%, the better. INE, on the other hand, directly reflects the size of the prediction set; an overly large prediction set implies high uncertainty. For single-label classification tasks, the optimal INE equals 1, meaning the prediction set contains only one label. If this label is the true label, COV equals 100%; otherwise, COV equals 0%.
> - Lines 461-464 indicate that using TUI alone may sometimes fail to achieve a good balance between COV and INE. In some cases, COV is satisfactory, but INE is too high, while in others, INE is close to 1, but COV does not meet expectations. The subsequent paragraph shows that CPAda further enhances the results achieved by TUI, addressing these issues effectively.

---

> ### Comment · Reviewer_qpy6 · 2024-11-27
>
> Thanks for your clarification.
>
> **A1:** If the plug-and-play approach is the main contribution, it is necessary to demonstrate its generalizability by integrating it with diverse CTTA methods. In addition, the results in the table for ViDA are relatively weak, making it difficult to conclude that TUI is effective.
>
> **A2:** The unfairness I am referring to is not between different baselines but rather between the cases with and without TUI. Since TUI relies on an additional dataset for calibration, the observed improvement in error (though marginal) might result from the use of the calibration dataset. Additionally, an ablation study on the size of the calibration dataset would provide valuable insights.
>
> **A3:**
>
> > Uncertainty estimation is only credible if the coverage gap (COV column) is addressed
>
> I believe uncertainty estimation is meaningful when comparing two models with similar accuracy. COV, however, seems irrelevant to uncertainty estimation in this context.
>
> Additionally, from my understanding, COV evaluates how many data points' true labels are included in the prediction set. A larger COV, which indicates a more accurate prediction set, should theoretically lead to a lower error rate. However, the results in Tables 1-3 do not support this assumption—despite a significant COV gap, the error and calibration improvements are only marginal. This raises the question: what is the value of a larger COV if it does not translate into meaningful improvements?
>
>
>
> > The use of softmax aims to project the concatenated results into the probability space, ensuring that each value in the output comes from a valid probability distribution.
>
> What is the meaning of the probability space when the same label appears twice in the predictions? How is this scenario connected to model shift? For instance, consider a 3-way classifier where the logits from the source model are $[1,1,1]$, but the logits from the current model are $[1000,1000,1000]$ (a simple scaling of weights by 1000). While the prediction vector remains the same for both models, the computed $p(x)$ through Eqn. 4 would be drastically different. How does this discrepancy reflect the underlying model shift, and what implications does it have for the method?

---

> ### Author Response · Authors · 2024-11-28
>
> Thanks for the further comments from the reviewer. The responses are as follows:
>
>
> **Response to Question 1:**
>
> We agree with the reviewer’s validation approach for the plug-and-play method. In fact, we have already validated it on three datasets using no fewer than five existing methods in the paper. As for the VIDA method mentioned, it shows good COV performance, which indicates that we can easily estimate which samples are uncertain within the VIDA method. For more details on estimating uncertainty, please refer to our response to the next question.
>
> **Response to Question 2:**
>
> The reviewer may have some misunderstandings in conformal prediction. In this paper, simply adding TUI does not change the original model's forward propagation or test results. In other words, there is no difference in the primary metric (ERR) when TUI is added versus when it is not. Adding TUI only provides an additional uncertainty estimation for the existing method, so there is no need to present a separate table showing the differences between the results with and without TUI. If it is necessary to include such a comparison, we suggest removing the columns for COV and INE results from the existing tables.
>
> Furthermore, the size difference of the calibration set is **already evaluated in Table 5** of the submitted paper. Experimental results indicate that a larger calibration set can improve COV, but there is no need to set a too-large calibration set.
>
> **Response to Question 3:**
>
> *About the COV and uncertainty estimation:*
>
> First, COV does not directly measure uncertainty, but it serves as a guarantee for uncertainty measurement. In fact, the size of INE (the number of labels in the prediction set) directly quantifies uncertainty. COV and INE are interdependent that sufficient COV is required for INE to be meaningful. To illustrate, consider a doctor’s diagnosis:
> - If the diagnosis is simply “pneumonia”, it does not account for uncertainty.
> - If the doctor states that there is a 90% probability (the required COV) that the condition is one of the following—[pneumonia, emphysema, pulmonary nodules] (INE=3)—this expresses the uncertainty of the diagnosis. The size of this set determines the level of uncertainty. If the doctor lists 10 possible conditions (INE=10), it shows greater uncertainty in the diagnosis. A smaller set suggests a more confident diagnosis.
> - If the doctor states that there is only a 30% probability (below the required COV) that the condition is one of the three possibilities (still INE=3), the diagnosis becomes unreliable.
>
> In the context of domain shift, directly using Conformal Prediction would result in a lower COV, making the uncertainty estimation untrustworthy, say the coverage gap. This paper proposes to solve the problem.
>
> The reviewer may refer to the following papers for more details on conformal prediction:
>
> 1. Shafer, Glenn, et al. "A tutorial on conformal prediction." Journal of Machine Learning Research 9.3 (2008).
> 1. Fontana, Matteo, et al. "Conformal prediction: a unified review of theory and new challenges." Bernoulli 29.1 (2023): 1-23.
> 1. Barber, Rina Foygel, et al. "Conformal prediction beyond exchangeability." The Annals of Statistics 51.2 (2023): 816-845.
> 1. Angelopoulos, Anastasios N., et al. "Conformal prediction: A gentle introduction." Foundations and Trends® in Machine Learning 16.4 (2023): 494-591.
>
>
> *About the joint logits:*
>
> In response to the reviewer’s question, when the same label appears twice in the predictions, it typically indicates that the model is confident in that label, possibly due to scaling or overemphasis in the logits. For example, if the logits from the source model are [1,1,1] and those from the current model are [1000,1000,1000], the predicted class remains the same, but the joint probabilities will differ significantly due to the scaling of logits. Despite the final prediction being identical, the underlying logit probability distributions are different, reflecting a shift in the model’s internal representation.
>
> It’s important to note that in our method, softmax in Eq 4 is not used as a classifier, but as a normalization technique to convert the logits into a unified probability space. This is used to capture distribution differences between models. The logits from the two models represent two different stages of model knowledge: one reflecting the past model's understanding and the other reflecting the current model’s understanding. These logits are spatially shifted—their distributions are different. If this shift is not acknowledged, there is essentially no model shift. To capture this difference, we apply the softmax (as in Eq. 4), which compresses the logits into a common probability space. The divergence between these distributions is then measured using JS divergence, which effectively captures both data and model shifts. This approach sensitizes to differences in how the models process data, even when their final predictions are the same.

---

> > ### Comment · Reviewer_qpy6 · 2024-11-29
> >
> > > **The unfairness I am referring to is not between different baselines but rather between the cases with and without TUI.**
> >
> > Let me clarify further. I am referring to the performance gap between cases with and without TUI+CPAda. For example, in Table 1, the comparison between the first row (Tent) and the second row (+CPAda).
> >
> > > **About the COV and uncertainty estimation:**
> >
> > Thanks for the authors' explanations regarding COV and INE. However, my question is about the statement: *"Uncertainty estimation is only credible if the coverage gap (COV column) is addressed."* Since metrics like NLL, BS, and ECE are computed using the true labels and predicted labels/confidence values, why are these metrics considered "not credible" when the COV is smaller, as seen in Table 1, first row (Tent)? Moreover, how do the authors explain the small gap in these metrics between cases with and without TUI+CPAda?

---

> > > ### Author Response · Authors · 2024-12-02
> > >
> > > ### About the unfairness between cases with and without TUI
> > > We appreciate the reviewer’s clarification of the concern. Our method leverages the Conformal Prediction framework, where COV (Coverage) measures the probability that the prediction set includes the true label, and INE (Prediction Set Size) represents the size of the prediction set. These two metrics together reflect the trade-off between prediction reliability and efficiency.
> > > The key contribution of TUI+CPAda lies in improving these two metrics under domain shifts. Without TUI+CPAda (e.g., the Tent baseline), the model operates without explicit coverage guarantees, leading to potential unreliability in prediction confidence, even if metrics like NLL, BS, and ECE show comparable values.
> > >
> > > ### Addressing COV and the credibility of uncertainty estimation
> > > We emphasize that the credibility of uncertainty estimation depends on achieving sufficient coverage (COV). While metrics such as NLL, BS, and ECE are computed using the predicted probabilities and true labels, they inherently assume that the predicted probabilities are reliable. When COV is lower, it implies that the prediction set fails to include the true label in some cases, indicating unreliability in the estimated probabilities. Therefore, even if NLL, BS, or ECE appear favorable, these metrics alone cannot confirm that the model provides trustworthy uncertainty estimates. By explicitly addressing the coverage gap, TUI+CPAda ensures that the uncertainty estimation aligns with the true reliability of the predictions.
> > >
> > > ### Explaining the Small Gap in Metrics
> > > We acknowledge the reviewer’s observation regarding the small differences in NLL, BS, and ECE between cases with and without TUI+CPAda. This phenomenon can be attributed to the inherent sensitivity of these metrics, which may not fully capture improvements in prediction reliability under domain shifts. TUI+CPAda’s primary contribution lies in improving COV, which ensures that predictions are robust and credible across varying domains. The small numerical gap in NLL and related metrics reflects their focus on average calibration and log-loss, whereas COV directly evaluates the fundamental reliability of the prediction set.

---

> > > > ### Comment · Reviewer_qpy6 · 2024-12-03
> > > >
> > > > > The unfairness I am referring to is not between different baselines but rather between the cases with and without TUI. Since TUI relies on an additional dataset for calibration, the observed improvement in error (though marginal) might result from the use of the calibration dataset.
> > > >
> > > > The authors misunderstood my question. The unfairness I am referring to is not between different baselines but rather between the cases with and without TUI+CPAda. Since TUI relies on an additional dataset for calibration, the observed **improvement in error rates** (though marginal) might result from the use of the calibration dataset.
> > > >
> > > > > While metrics such as NLL, BS, and ECE are computed using the predicted probabilities and true labels, they inherently assume that the predicted probabilities are reliable.
> > > >
> > > > What is the definition of "reliability of predicted probabilities"? Are you suggesting that only correct predictions have reliable probabilities? Any reference supporting this statement would be helpful. My understanding is that higher COV indicates that the true labels are included in the prediction set. Does this imply that if top-k accuracy were computed, the proposed method might show greater improvement over the baselines? Providing such results could be helpful. That said, the marginal gap in uncertainty metrics remains confusing and raises questions about the method's effectiveness.
> > > >
> > > > Finally, I would like to note that the sluggish rebuttal process has left insufficient time for further discussion. As a result, I will maintain my current rating.

---

### Official Review · Reviewer_k3DK · 2024-11-02

**Soundness:** 2
**Presentation:** 3
**Contribution:** 3
**Rating:** 5
**Confidence:** 3

**Summary:**

The paper introduces the Test Uncertainty Indicator (TUI), a novel method for Continual Test-Time Adaptation (CTTA) that estimates uncertainty in test samples within dynamically changing environments. By leveraging Conformal Prediction (CP), TUI generates prediction sets with a specified coverage probability, enabling models to select genuinely high-confidence samples for adaptation. The authors propose a quantile compensation strategy to address coverage gaps arising from domain shifts, effectively reducing these gaps. Extensive experiments on CIFAR10-C, CIFAR100-C, and ImageNet-C demonstrate that TUI surpasses traditional CP methods in handling uncertainty and adapting to shifting domains.

**Strengths:**

1. **Clear Problem Identification**: The paper addresses a critical CTTA challenge—error accumulation from pseudo-labeling under domain shifts. It underscores the importance of accurate uncertainty estimation, particularly as most TTA methods rely on self-training.

2. **Effective Uncertainty Estimation with CP**: TUI uses Conformal Prediction to ensure each prediction meets a specified coverage probability. By dynamically adjusting thresholds for domain shifts, it provides a more adaptive and suitable approach for TTA.

3. **Broad Applicability to TTA Methods**: The proposed CPAda uses high-confidence samples for sample-wise adaptation, serving as a versatile, plug-and-play enhancement to various TTA methods, boosting robustness and accuracy.

**Weaknesses:**

1. **Modest Performance Gain on Error Rate**: While TUI enhances coverage and robustness, its impact on error rate is modest (often less than 1%), which is a crucial metric for TTA. The added complexity and data requirements may not justify these minor gains for some applications.

2. **Computational Overhead**: The additional steps in TUI and CPAda, particularly dynamic quantile compensation and uncertainty estimation, increase computational and memory demands, potentially hindering real-time use. Comparative experiments on computational and memory overhead with and without CPAda would strengthen the evaluation.

3. **Lack of Ablation Study**: The paper does not include an ablation study to isolate the effects of individual TUI components, such as dynamic thresholding, making it difficult to understand the specific impact of each on performance.

4. **Dependency on Additional Calibration Set**: TUI requires an extra calibration set, which may be impractical in real-world applications due to storage limits or privacy constraints. Additionally, it remains unclear if CPAda would outperform methods like EATA[1] and RMT[2] that leverage additional source domain data.

[1] Niu, Shuaicheng, et al. "Efficient test-time model adaptation without forgetting." International conference on machine learning. PMLR, 2022.

[2] Döbler, Mario, Robert A. Marsden, and Bin Yang. "Robust mean teacher for continual and gradual test-time adaptation." Proceedings of the IEEE/CVF Conference on Computer Vision and Pattern Recognition. 2023.

**Questions:**

Please refer to **Weaknesses**.

---

> ### Author Response · Authors · 2024-11-25
>
> ## A1
>
> TUI is a method for evaluating uncertainty, and directly incorporating TUI does not alter the original model's training outcomes. Guided by TUI, it can somewhat reduce the model's uncertainty and slightly improve its performance. The method we proposed is an insurance method, not an acceleration method. In the testing scenario, our method has two advantages: plug-and-play and model-agnostic. The proposed method is particularly suitable for scenarios with long-term testing, where the consequences of error accumulation are unacceptable (e.g., autonomous driving or medical applications) and where early knowledge of the model's uncertainty is crucial. It is not feasible to intervene manually only after a medical mishap or to adjust the model after a traffic accident. Being able to anticipate the uncertainty of results in advance is beneficial for subsequent operations.
>
> ## A2
> Sure, the calibration process requires additional computation for the calibration data, but it is small compared to the vanilla methods. Specifically, the time (s/batch) of the process can be calculated  in the following table:
> | alpha=0.2      | cotta  | rmt    | ccotta | ViDA    |
> |----------------|--------|--------|--------|---------|
> | NONE           | 1.0695 | 1.0307 | 1.1262 | 1.2542  |
> | NONE+TUI       | 1.2097 | 1.1643 | 1.2602 | 1.2599  |
> | NONE+TUI+CPAda | 1.2118 | 1.2155 | 1.2807 | 1.2904  |
>
> The method we proposed is **an insurance method**, not an acceleration method. In any testing scenario, our method has two advantages: **plug-and-play** and **model-agnostic**. Our method is more suitable for long-term testing **scenarios where errors is unacceptable** (e.g., autonomous driving, medical scenarios), and it is necessary to know the model's uncertainty in advance. We cannot intervene after a medical incident, nor can we correct the model after a traffic accident. If we can know the uncertainty of the results in advance, it will be helpful for subsequent operations.
>
> ## A3
>
> The proposed method consists of only two components: TUI and TUI-based CPAda. All the comparison methods in this paper (Tables 1, 2, and 3) incorporate TUI, and the results using CPAda are provided in the subsequent rows. Therefore, the tables in the manuscript already present the results for the two main components, and there is no need to include an additional ablation study table.
>
> ## A4
>
> The main objective of this paper is to build upon existing CTTA methods by leveraging a small calibration set to evaluate the reliability of upcoming test results and use this evaluation to improve the certainty of adaptation. The proposed method calibrates the model using the calibration set, which can be either extracted from the training set or constructed separately, making it entirely independent of the model.
>
> In our manuscript, we have compared our method with the source replay approach used in RMT, as shown in Table 5 of the paper. The results demonstrate that our method outperforms directly using the source replay approach.

---

> > ### Comment · Reviewer_k3DK · 2024-12-02
> >
> > Thank you for your response. I appreciate the clarifications provided, but I still believe the paper needs further improvements before it can be considered for acceptance. Specifically, while there are some marginal improvements, additional supporting data is required. Moreover, as an off-the-shelf method, it hasn't been evaluated using various TTA methods. Therefore, I plan to maintain my initial score.

---

### Official Review · Reviewer_45KG · 2024-11-03

**Soundness:** 3
**Presentation:** 3
**Contribution:** 3
**Rating:** 6
**Confidence:** 3

**Summary:**

This paper introduces a novel uncertainty indicator called Test Uncertainty Indicator (TUI) for continuous test time adaptation setting. Based on the existing research in Conformal Prediction Task, the authors point out that the key challenge in CTTA is the continuously changing test domain, which is neglected by the traditional CP methods. In this paper, the authors propose to compute the joint domain shifts and compensate the quantile threshold to obtain the prediction set of the test samples, which is then used to measure the uncertainty. With the help of TUI, several state-of-the-art TTA methods have been demonstrated to achieve significant improvement and better calibration performance on CIFAR10/100-C and ImageNet-C benchmark datasets.

**Strengths:**

1. The paper is well written and easy to follow. The authors go from introducing the traditional CP methods to the Coverage Gap Issue on CTTA and then introduce a novel solution named TUI. The whole line of thought is very coherent and easy to understand.
2. The evaluation experiments are sufficient and are conducted on several of the most representative CTTA datasets.

**Weaknesses:**

1. The proposed TUI method is necessary to maintain a calibration set consisting of tens or hundreds of source samples, which is a sensitive assumption under the CTTA setting. According to the definition of TTA / CTTA in TENT[A] and CoTTA[B], the source data cannot normally be used due to the privacy issue.
2. The hyper-parameters, e.g. $\beta$, $\alpha$, are introduced in TUI. The authors should perform sensitivity analyses on the introduced hyper-parameter $\beta$ and indicate how to decide the hyper-parameters while deploying the algorithm in new testing dataset.

[A] Tent: Fully Test-time Adaptation by Entropy Minimization
[B] Continual Test-Time Domain Adaptation

**Questions:**

1. Does the notation $\pi(x)$ represent the probabilities from pre-trained model? Since $\pi_{\theta^{src}}(x)$ and $\pi_{\theta^{crt}}(x)$ appearing on the Eq. 4 represent the probabilities from source model and current model, respectively, it's not clear to me which model is used by the presence of $\pi(x)$ alone in equation 7.

---

> ### Author Response · Authors · 2024-11-25
>
> ## A1:
> Our paper is plug-and-play, and builds upon existing Continual Test-Time Adaptation (CTTA) methods, aiming to assess the credibility of upcoming test results with just a small set of calibration data and to improve the adaptation effect using this assessment. Our approach calibrates the model using a calibration set, which can be derived from the training set or constructed separately, and is entirely independent of the model itself.
> - **Fairness**: We have considered that previous methods used all training data during pre-training. Therefore, all comparative methods (as shown in all experimental charts and tables) in our paper employ calibration sets constructed in the same manner for conformal prediction, ensuring there is no issue of unfairness.
> - **Privacy**: The construction of the calibration set can be independent of the training set, serving merely as an additional reference to assist in measuring the uncertainty of test results during testing. To illustrate this, we separate the source training data into training and calibration datasets during the pre-training phase. We retrained the pre-trained model only on the separated training data, without training on the calibration data, thus eliminating privacy concerns. The results in this scenario are as follows:
>
> | With seperated calibration set          | ERR   | COV   | INE  | NLL  | BS   | ECE   |
> |-----------------|-------|-------|------|------|------|-------|
> | tent+TUI        | 19.01 | 70.68 | 0.84 | 1.67 | 0.16 | 0.16  |
> | tent+TUI+CPAda  | 18.89 | 77.76 | 1.06 | 1.75 | 0.17 | 0.19  |
> | CoTTA+TUI       | 17.44 | 83.83 | 1.68 | 0.57 | 0.11 | 0.05  |
> | CoTTA+TUI+CPAda | 16.61 | 78.71 | 1.11 | 0.64 | 0.12 | 0.12  |
> | SATA+TUI        | 16.65 | 47.46 | 0.51 | 0.57 | 0.1  | 0.03  |
> | SATA+TUI+CPAda  | 16.51 | 77.6  | 0.98 | 0.67 | 0.11 | 0.1   |
> | RMT+TUI         | 17.36 | 88.35 | 1.88 | 0.69 | 0.12 | 0.05  |
> | RMT+TUI+CPAda   | 16.7  | 78.56 | 0.98 | 0.75 | 0.12 | 0.14  |
>
> - **Other Methods Utilizing Source Data**: We also note that other methods, such as RMT, leverage source data in their approach (Source replay). We have compared with source replay in our manuscript (Table 5), and found that our CP outperforms replay-based methods.
>
> ## A2
>
> Regarding the two hyperparameters mentioned by the reviewers, we would like to provide clarification:
>
> - **Hyperparameter α**: This parameter is specified by the user in Conformal Prediction as the acceptable error rate. For instance, if α is set to 0.1, the resulting prediction set is expected to achieve a 90% coverage rate. In other words, there is a 90% probability that the true label will be included within the prediction set. A similar explanation can be found in Equation (2). In most of our experiments, we set α to three levels: 0.1, 0.2, and 0.3.
> - **Hyperparameter β**: This parameter is a constant used to control the magnitude of the compensation term ρ. Since ρ is generally small, β is typically set to a relatively large value. The following table presents the experimental results for hyperparameter β on CIFAR10-C, C-CoTTA, and α=0.2. Note that a larger β may make the coverage to 100% and inefficiency to10, which means a full prediction set (all labels included). The CP is not working. When β is small, the coverage has a gap to the expectation (α=0.2, the expectation is 80%), and the uncertainty measure is not reliable.
>
> |   β   | ERR   | COV   | INE  | NLL  | BS   | ECE   |
> |------|-------|-------|------|------|------|-------|
> | 15e4 | 14.97 | 58.12 | 0.6  | 0.88 | 0.12 | 0.16  |
> | 15e5 | 14.98 | 58.37 | 0.6  | 0.88 | 0.12 | 0.17  |
> | 15e6 | 14.96 | 59.3  | 0.61 | 0.88 | 0.12 | 0.15  |
> | 15e7 | 14.88 | 60.18 | 0.63 | 0.88 | 0.12 | 0.15  |
> | 15e8 | 14.88 | 73.76 | 0.81 | 0.9  | 0.12 | 0.16  |
> | 15e9 | 15.04 | 100   | 10   | 0.86 | 0.12 | 0.13  |
>
> ## A3 to the Question
>
> The reviewer's understanding of Eq. (4) is correct. In Eq. (7), $ \pi(x) $ refers to the current model. In this version, we will clarify this description to avoid confusion.

---

### Official Review · Reviewer_wwYx · 2024-11-04

**Soundness:** 1
**Presentation:** 2
**Contribution:** 2
**Rating:** 3
**Confidence:** 4

**Summary:**

This paper aims to mitigate error accumulation during test-time adaptation (TTA) by reducing the impact of model adaptation when the model prediction is deemed too uncertain. To achieve this, the authors propose storing a portion of the labeled source dataset as a calibration set and introducing TUI, an uncertainty indicator method based on conformal prediction. TUI weights the influence on the training process for each test sample based on this uncertainty.

**Strengths:**

The proposed method is model-agnostic and can be applied orthogonally to existing CTTA approaches.

**Weaknesses:**

[W1] Concerns arise regarding the practice of storing and utilizing source data. TTA research operates under realistic assumptions, with related studies avoiding access to training data during testing. In scenarios where training data cannot be accessed due to data privacy issues, how can this limitation be addressed? The paper "Leveraging Proxy of Training Data for Test-Time Adaptation, ICML 2023" suggests utilizing condensed data to leverage training set information while maintaining data privacy in TTA contexts. Could TUI also be applied when employing a proxy for training data, similar to the aforementioned approach?

[W2] Regarding complexity, the use of both a pretrained source model and the testing model may significantly increase time complexity. Additionally, the requirement to forward both the test and calibration data for each sample adds to this complexity. How much time does complexity increase as a result?

[W3] The corruption benchmark focuses on synthetic domain shifts, which may not reflect realistic scenarios. Validation of the method’s effectiveness on natural domain shift benchmarks is necessary. Does it perform well on natural domain shift benchmarks like VisDA-C and Cityscape-to-ACDC, which have been utilized in prior CTTA research?

[W4] Furthermore, there appears to be insufficient comparison with recent literature. Could the authors provide a comparison with the following works?

- (SAR) Towards stable test-time adaptation in dynamic wild world, ICLR 2023

- (DePT) Visual prompt tuning for test-time domain adaptation

- (VDP) Decorate the newcomers: Visual domain prompt for continual test-time adaptation, AAAI 2023

- (EcoTTA) Ecotta: Memory-efficient continual test-time adaptation via self-distilled regularization, CVPR 2023

- (SVDP) Exploring sparse visual prompt for cross-domain semantic segmentation, AAAI 2024

- Reshaping the online data buffering and organizing mechanism for continual test-time adaptation, ECCV 2024

- A versatile framework for continual test-time domain adaptation: Balancing discriminability and generalizability, CVPR 2024

- (Continual-MAE) Continual-MAE: Adaptive distribution masked autoencoders for continual test-time adaptation, CVPR 2024

- (ViDA) Vida: Homeostatic visual domain adapter for continual test-time adaptation, ICLR 2024

- (BECoTTA) BECoTTA: Input-dependent Online Blending of Experts for Continual Test-time Adaptation, ICML 2024

[W5] Using labeled source data may be perceived as somewhat akin to cheating. Given this, is the observed performance marginal? Is the difference in performance statistically significant?

**Questions:**

It is unclear whether the calibration set is constructed as a subset of the training set or if it is derived from the validation set of the source dataset. Could the authors clarify which approach is used?

---

> ### Author Response · Authors · 2024-11-25
>
> ## A1:
> Our paper is plug-and-play, and builds upon existing Continual Test-Time Adaptation (CTTA) methods, aiming to assess the credibility of upcoming test results with just a small set of calibration data and to improve the adaptation effect using this assessment. Our approach calibrates the model using a calibration set, which can be derived from the training set or constructed separately, and is entirely independent of the model itself.
> - **Fairness**: We have considered that previous methods used all training data during pre-training. Therefore, all comparative methods (as shown in all experimental charts and tables) in our paper employ calibration sets constructed in the same manner for conformal prediction, ensuring there is no issue of unfairness.
> - **Privacy**: The construction of the calibration set can be independent of the training set, serving merely as an additional reference to assist in measuring the uncertainty of test results during testing. To illustrate this, we separate the source training data into training and calibration datasets during the pre-training phase. We retrained the pre-trained model only on the separated training data, without training on the calibration data, thus eliminating privacy concerns. The results in this scenario are as follows:
>
> | With seperated calibration set          | ERR   | COV   | INE  | NLL  | BS   | ECE   |
> |-----------------|-------|-------|------|------|------|-------|
> | tent+TUI        | 19.01 | 70.68 | 0.84 | 1.67 | 0.16 | 0.16  |
> | tent+TUI+CPAda  | 18.89 | 77.76 | 1.06 | 1.75 | 0.17 | 0.19  |
> | CoTTA+TUI       | 17.44 | 83.83 | 1.68 | 0.57 | 0.11 | 0.05  |
> | CoTTA+TUI+CPAda | 16.61 | 78.71 | 1.11 | 0.64 | 0.12 | 0.12  |
> | SATA+TUI        | 16.65 | 47.46 | 0.51 | 0.57 | 0.1  | 0.03  |
> | SATA+TUI+CPAda  | 16.51 | 77.6  | 0.98 | 0.67 | 0.11 | 0.1   |
> | RMT+TUI         | 17.36 | 88.35 | 1.88 | 0.69 | 0.12 | 0.05  |
> | RMT+TUI+CPAda   | 16.7  | 78.56 | 0.98 | 0.75 | 0.12 | 0.14  |
>
> - **Other Methods Utilizing Source Data**: We also note that other methods, such as RMT, leverage source data in their approach (Source replay). We have compared with source replay in our manuscript (Table 5), and found that our CP outperforms replay-based methods.
>
> ## A2:
> Sure, the calibration process requires additional computation for the calibration data, but which is small compared to the vanilla methods. Specifically, the time (s/batch) of the process can be calculated as follows:
> | alpha=0.2      | cotta  | rmt    | ccotta | ViDA    |
> |----------------|--------|--------|--------|---------|
> | NONE           | 1.0695 | 1.0307 | 1.1262 | 1.2542  |
> | NONE+TUI       | 1.2097 | 1.1643 | 1.2602 | 1.2599  |
> | NONE+TUI+CPAda | 1.2118 | 1.2155 | 1.2807 | 1.2904  |
>
> The method we proposed is **an insurance method**, not an acceleration method. In any testing scenario, our method has two advantages: **plug-and-play** and **model-agnostic**. Our method is more suitable for long-term testing **scenarios where errors is unacceptable** (e.g., autonomous driving, medical scenarios), and it is necessary to know the model's uncertainty in advance. We cannot intervene after a medical incident, nor can we correct the model after a traffic accident. If we can know the uncertainty of the results in advance, it will be helpful for subsequent operations.
>
> ## A4:
>
> - The objective of this paper is not to compete with the latest SOTA CTTA methods but to propose a **plug-and-play approach** for measuring the uncertainty of test results in existing methods, primarily assessing whether it can effectively address the coverage gap in conformal prediction. As a plug-and-play method, our approach has been constructed on top of some existing CTTA methods (Table 1,2,3), and the results demonstrate its effectiveness.
> - Our method designs a conformal prediction technique, and we **have compared it with the most SOTA conformal prediction methods** including NexCP and QTC. The comparative results are presented in Table 4.
> - To address the reviewer's concerns, we select additional ViDA to evaluate the uncertainty measurement outcomes, and the results are as follows:
> | Method     | ERR   | COV   | INE  | NLL  | BS   | ECE   |
> |------------|-------|-------|------|------|------|-------|
> | $\alpha = 0.3$        |       |       |      |      |      |       |
> | vida+TUI       | 18.64 | 44.11 | 0.48 | 0.94 | 0.14 | 0.14  |
> | vida+TUI+CPAda | 18.61 | 69.49 | 2.72 | 0.95 | 0.15 | 0.16  |
> | $\alpha = 0.2$        |       |       |      |      |      |       |
> | vida+TUI       | 18.64 | 54.29 | 0.65 | 0.94 | 0.14 | 0.14  |
> | vida+TUI+CPAda | 18.63 | 76.23 | 2.86 | 0.95 | 0.15 | 0.17  |
> | $\alpha = 0.1$        |       |       |      |      |      |       |
> | vida+TUI       | 18.64 | 67.45 | 1    | 0.94 | 0.14 | 0.14  |
> | vida+TUI+CPAda | 18.53 | 85.29 | 3.27 | 0.95 | 0.15 | 0.17  |

---

> ### Author Response · Authors · 2024-11-25
>
> ## A3:
>
> Thank you for your insightful comments regarding the datasets mentioned. We would like to address your concerns with the following clarifications and additional information:
> - **VisDA-C Dataset**: You are correct in noting that VisDA-C is a classic dataset for Unsupervised Domain Adaptation (UDA) and does not inherently involve the multiple domain shifts present in CTTA scenarios, nor does it exhibit error accumulation. We will provide results in this scenario to further validate our method's effectiveness across different settings.
> - **Cityscapes-to-ACDC Dataset**: This dataset represents a segmentation task, which is more aligned with the continuous distribution changes in real-world scenarios. Our proposed method is designed to estimate uncertainty at the image level, the application of Conformal Prediction (CP) for pixel-level uncertainty estimation is an unexplored area. We acknowledge the novelty and potential of extending our approach to pixel-level uncertainty estimation, which could be a valuable direction for future research.
> - Many existing methods have employed the datasets used in this paper to validate the effectiveness of their methods at the image level, including CoTTA, RMT, ViDA.

---

### Note · Authors · 2025-01-15

I have read and agree with the venue's withdrawal policy on behalf of myself and my co-authors.